# MXene as Promising Anode Material for High-Performance Lithium-Ion Batteries: A Comprehensive Review

**DOI:** 10.3390/nano14070616

**Published:** 2024-03-31

**Authors:** Mohammad Nezam Uddin Chy, Md. Arafat Rahman, Jin-Hyuk Kim, Nirjhor Barua, Wasif Abu Dujana

**Affiliations:** 1Department of Mechanical Engineering, Chittagong University of Engineering & Technology, Chittagong 4349, Bangladesh; nezam.uddin@cuet.ac.bd (M.N.U.C.); nirjhorbarua371@gmail.com (N.B.); 2Carbon Neutral Technology R&D Department, Korea Institute of Industrial Technology, Cheonan 31056, Republic of Korea; 3Convergence Manufacturing System Engineering (Green Process and Energy System Engineering), University of Science & Technology, Daejeon 34113, Republic of Korea; 4Department of Materials and Metallurgical Engineering, Chittagong University of Engineering & Technology, Chittagong 4349, Bangladesh; wasif@cuet.ac.bd

**Keywords:** MAX phase, MXene, anode, capacity, lithium-ion battery

## Abstract

Broad adoption has already been started of MXene materials in various energy storage technologies, such as super-capacitors and batteries, due to the increasing versatility of the preparation methods, as well as the ongoing discovery of new members. The essential requirements for an excellent anode material for lithium-ion batteries (LIBs) are high safety, minimal volume expansion during the lithiation/de-lithiation process, high cyclic stability, and high Li^+^ storage capability. However, most of the anode materials for LIBs, such as graphite, SnO_2_, Si, Al, and Li_4_Ti_5_O_12_, have at least one issue. Hence, creating novel anode materials continues to be difficult. To date, a few MXenes have been investigated experimentally as anodes of LIBs due to their distinct active voltage windows, large power capabilities, and longer cyclic life. The objective of this review paper is to provide an overview of the synthesis and characterization characteristics of the MXenes as anode materials of LIBs, including their discharge/charge capacity, rate performance, and cycle ability. In addition, a summary of the potential outlook for developments of these materials as anodes is provided.

## 1. Introduction

After extensive investigation and usage, consumers are acquainted with lithium-ion batteries. They are now frequently utilized in electrical appliances and electric automobiles [1,2]. A potential market for electric urban automobiles has developed recently, with producers of mobile devices engaging in a competitive series. Advancements in high-performance lithium-ion batteries, such as development of high specific capacity, energy density, high open circuit voltage, and low self-discharge, are being prioritized [3,4,5]. MXenes have potential in a variety of energy storage applications due to their improved electrical conductivity, outstanding mechanical strength, and immense surface area. A MXene has the chemical formula M_n+1_X_n_T_x_, where M is a transition metal (i.e., Sc, Ti, V, Zr, Nb, Mo, etc.), X is C or N, and T denotes surface terminal groups (i.e., –O, –OH, and –F). The performance of MXenes in terms of capacitive energy storage is quite encouraging. Ti_3_C_2_T_x_, for example, has demonstrated excellent volumetric and gravimetric capacitance, as well as outstanding cyclability [6]. In the aqueous H_2_SO_4_ electrolyte, MXene displays pseudocapacitance due to the reversible surface redox reaction of hydrogen binding. The electric double-layer (EDL) capacitance of MXene is predominant in aqueous salt or ionic liquid electrolytes [7]. When two transition metals are coupled in an MXene structure, a special property of MXenes is triggered into action. Transition metals can form ordered structures in a single 2D MXene flake in addition to the expected solid solutions, such as (Ti,Nb)CT_x_, by forming either atomic sandwiches of transition metals planes (for *n* ≥ 2) such as Mo_2_TiC_2_T_x_, or in-plane (*n* = 1) ordered structures such as (Mo_2/3_Y_1/3_)_2_CT_x_. A large number of potential compositions have been anticipated from computational investigations of MXenes and their predecessors. The synthesis of an endless number of nonstoichiometric MXenes is possible with the creation of solid solutions on M and/or X sites, and this also presents an alluring chance of optimizing characteristics by combining other transition metals or synthesizing carbonitrides. There is continuing research to generate 2D borides and so provide the system with another X component [8,9].

Recently, an electrochemical fluoride-free production technique utilizing dilute hydrochloric acid was disclosed for Ti_2_CT_x_ MXene production [10]. Huang et al. created novel MAX phases and fluorine-free MXenes using molten ZnCl_2_ salt [11]. For those people who are interested in MXenes but do not wish to operate in laboratories with any chemicals that include or produce hydrofluoric acid (HF), this method has the potential to considerably expand experimental research on MXenes. A successful use of V_2_CT_x_ MXene as a positive electrode material for sodium-ion storage has already been developed [12]. It is noted that porous MXene-based paper electrodes with large volumetric capacities and reliable cycling performance are promising for sodium-based energy storage systems when the size is considerable [13]. p-Ti_3_C_2_T_x_/CNT electrodes demonstrated significantly superior Li-ion storage performance compared to non-porous Ti_3_C_2_T_x_/CNT films [14]. Ti_3_C_2_T_x_, the first MXene to be identified, has been the subject of more than 70% of all MXene studies. However, because of how thoroughly this MXene has been investigated, a large number of researchers now simply refer to it as Ti_3_C_2_T_x_ when referring to MXene, which can be deceptive because MXenes can have a broad range of structural variations and chemical compositions. At least 100 stoichiometric MXene compositions and an unlimited number of solid solutions not only provide uncommon combinations of characteristics, but also the opportunity to customize such traits by varying the ratios of the M or X components. Therefore, considering the numerous possible compositions of MXenes, and the encouraging experimental findings in energy storage and reliable cycling, it is worth investigating different compositions of MXenes as anode materials of LIBs.

It is worth mentioning that the exceptional properties, including good electronic conductivity, low operating voltage, fast Li^+^ diffusion, and high theoretical Li storage capacity, suggest MXenes’ potential as materials of energy storage devices in the future. However, most of these characteristics are from theoretical or computational studies. Hence, it is necessary to experimentally study more extensively how these materials would behave in energy storage devices. In this review, we comprehensively report on various experimental studies of MXene materials as anodes of LIBs, and also outline future research trends.

## 2. Structural Framework of MXene

This research topic started with the report of ‘Al’ extraction from the MAX phase Ti_3_AlC_2_, in which a 2D material Ti_3_C_2_ was formed having hydroxyl (OH) and/or fluoride (F) as surface groups [15]. Since then, a tremendous number of studies has been performed with MXene structures, and the numbers of articles for MXene are increasing exponentially. The MAX phase has the general formula of M_n+1_AX_n_ in which ‘M’ and ‘A’ symbolize transition metals of the d-block and elements primarily from the p-block, respectively. The ‘X’ in the formula represents N or C atoms, or both [16]. The mother MAX phase typically undergoes etching to eliminate the ‘A’ layer of atoms, and after sonication the MXene structure is obtained. The overall procedure of synthesizing MXene from the MAX phase is expounded upon in another section within this paper.

A MXene structure can consist of multiple layers of the constituent atoms. When the number of layers in a MXene is less than 5, it can be called a few-layer MXene [17]. A single layer of an M_2_X MXene structure has M-X-M layers of atoms, which means the X atom layer is sandwiched between two M atom layers. These layers of M and X atoms will create M_6_X octahedrons, as seen in Figure 1. In addition, the structure is hexagonal-shaped, having an ABABAB stacking sequence. This AB-type stacking can be observed for Ti_3_C_2_T_x_ [18] and Ti_3_C_2_Cl_x_ [11], where the terminations contribute to form octahedrons. It is noted that, for AB stacking, triangular prisms can be obtained between the layers rather than octahedrons. By etching the MAX phase with HF, some Ti_3_C_2_T_x_ structures can have a type of stacking sequence [19]. However, studies exhibited that there can also be a mixing of these two types of stacking sequences [20]. Furthermore, M_3_X_2_ and M_4_X_3_ structures sometimes prefer to have an ABCABC stacking sequence [21].

A surface of MXene layers will have functional groups referred to as surface terminating groups. Mixed terminating groups, such as =O, –OH, –Cl, and –F, can be generally seen to cover the surfaces of MXenes [22]. These functional groups are the result of the etching process undertaken to synthesize MXene structures. Experimentally, these terminating groups can be randomly distributed over the surface of MXenes. For instance, the as-prepared sample of Ti_3_C_2_T_x_ has the highest −F terminating group on its surface, whereas the fraction of –O terminations is around 0.3. However, aging of the sample may double the fraction of –O terminating groups at the expense of –F terminating groups [23].

It is noted that –O and/or –OH termination groups are the most stable since −F termination groups can easily be replaced with –OH groups while being stored in water [24]. MXenes terminated fully with oxygen are thermodynamically more stable than partially terminated structures [25]. However, –OH groups can be transformed to –O groups when the sample is treated with high temperatures [24]. The sites to which the terminating groups are attached is another important factor when considering stability. For instance, structures of Ti_2_C(OH)_2_ and Ti_3_C_2_(OH)_2_ are more stable when the –OH groups hold the position of hollow sites having three carbon atoms as neighbors [26]. Sometimes, inserting a transition element into the MXene structure can prevent the formation of the unstable M-X bond. In this way, the Mo–C bonds in Mo_3_C_2_T_x_ and Mo_4_C_3_T_x_ are suppressed by adding Ti into the structure, which makes it more stable [27,28].

However, it is important to modify these functional groups to explore the remarkable properties of MXenes [12,29]. The O/OH terminating groups can aid the transportation of charges [30]. Methoxy terminated MXenes can be used for the esterification process as catalysts [31]. Transmittance can become high when a structure has –F or –OH terminating groups [32]. In addition, the weak bonds between the MX layers can lead to intercalation and thereby increase the layer spacing (along the c axis) of the MXene structure [33]. Intercalation with different metallic ions promotes usage of MXenes in metal ion energy storage systems [34]. Capacitors with superior capacitance can be constructed using these cation intercalated MXenes, the properties of which are equivalent to activated graphene [35]. However, a tremendous increase in capacitance has been reported with the use of K^+^ ion intercalation, with a value of 517 Fg^−1^ in Ti_3_C_2_T_x_ after removing the terminating groups with calcination [36]. Furthermore, not only can cations be intercalated into MXenes, but also organic molecules. For instance, a previous report showed the independent increase in the lattice parameter of Ti_3_C_2_ by the intercalation of organic hydrazine, dimethyl-sulfoxide (DMSO), and urea [33]. A increased volumetric capacitance of 1873 F cm^−3^ was found when a Ti_3_C_2_ MXene was intercalated with organic N,N dimethylacetamide [37].

## 3. Synthesis of MXene

MAX is a bulk crystal used in the synthesis of an MXene, which is itself a two-dimensional inorganic substance. It is noted that 2D layered materials produced from the MAX or non-MAX phase would exist before discovery. Unlike most 2D ceramics, MXenes, which are molecular sheets made from the carbides and nitrides of transition metals such as titanium, have exceptional conductivity and great volumetric capacitance by nature. However, more than 100 MXene compounds have been reported and several others have undergone computational investigations since the discovery of the first MXene (Ti_3_C_2_T_x_). In addition, MXenes may potentially consist of millions of different combinations of carbon, nitrogen, and transition metals such as molybdenum or titanium. It is noted that MXenes are produced by selectively eliminating aluminum from stacked MAX phases. The carbide layers are exfoliated into two sheets of MXene that are only a couple of atoms thick. MXenes can accommodate other ions and molecules between their layers by a procedure called intercalation, which is occasionally required to make use of the material’s special features. Inserting lithium ions between the sheets of MXene could transform them into effective materials for LIBs and capacitors. HF etchant, which initially was hazardous, has been supplanted with safer electrochemical etching as etching techniques have evolved over time [38]. In this context, the employment of various etchants often results in variations in surface terminations, which should further alter the structural, electrical, and chemical characteristics directly connected to the performance of energy storage devices. The three most common etching techniques are discussed briefly in this study.

### 3.1. Wet Chemical Etching

Ti_3_AlC_2_ MAX ceramic served as the precursor for the first MXene, Ti_3_C_2_T_X_, which was produced using HF solution as an etchant, as shown in Figure 2.

Nb_2_CT_X_, Ti_3_CNT_X_, and V_2_CT_X_ were among the other numerous MXenes that were produced using this method. The suggested etching procedure and related mechanism for Al-based MAX might be written as the following equations [42]:M_n+1_AlX_n_ + 3HF → M_n+1_X_n_ + AlF_3_ + 1.5H_2_
(1)
M_n+1_X_n_ + 2H_2_O → M_n+1_X_n_(OH)_2_ + H_2_
(2)
M_n+1_Xn + 2HF → M_n+1_X_n_F_2_ + H_2_
(3)

The metallic M-A bonds, which are considerably weaker than M-X (covalent or ionic bonds), break first, as illustrated in Equation (1), and F ions subsequently join with the Al ions to create AlF_3_, with the generation and release of H_2_ [43]. The Al layer is gradually removed from MAX, and the hexagonal lattice is transferred to MXene. Due to its high activity at this point and inability to be stable in either water or acid, MXene spontaneously reacts with H_2_O and HF, limiting surface energy by producing –F, =O, and –OH surface terminations.

### 3.2. Molten Salt Etching

In the past few years, molten salt etching has expanded into a novel technique that primarily targets nitride MXenes, as shown in Figure 3a. Ti_4_AlN_3_ MAX was first combined with the etchant (a combination of NaF, KF, and LiF), sintered at 550 °C for 30 min with flowing Ar, and the reaction byproducts were then eliminated using a 4 M H_2_SO_4_ solution for 1 h [44]. However, the resulting few-layered Ti_4_N_3_T_X_ MXene had more surface flaws than its HF-etched competitors in terms of crystallinity. This process is far more complex than wet chemistry. However, it is occasionally necessary due to the high temperature, complex fluoride salts, inert environment, and poisonous acid solution that are needed.

A comprehensive approach was suggested to create MXenes with halides at the end using Zn-based MAX as a precursor and merely ZnCl_2_ salts as the etchant [11]. After being heated at 550 °C for 5 h in an environment containing a solely Ar atmosphere, a group of MXenes, including Ti_3_C_2_Cl_2_ and Ti_2_CCl_2_, were created for the first time. The following equations illustrate the associated process that was elucidated in the replacement reaction between MAX ceramics and the late-transition metal halides [42]:Ti_3_ZnC_2_ + ZnCl_2_ → Ti_3_C_2_Cl_2_ + 2Zn^2+^(4)
Ti_3_ZnC_2_ + Zn^2+^ → Ti_3_C_2_ + 2Zn^2+^(5)
Ti_3_C_2_ + 2Cl^−^ → Ti_3_C_2_Cl_2_ + 2e^−^(6)
2Zn^2+^ + 2e^−^ = 2Zn(7)

It is noted that despite the significant advancement, the aforementioned inherent limitation is likely to render the molten salt etching approach less practical than wet chemical etching for the time being.

### 3.3. Electrochemical Etching

In order to etch the Ti_3_AlC_2_ MAX and produce the Ti_3_C_2_T_X_ MXene with –OH and =O terminations, Feng et al. [45] used a binary electrolyte, which is 1 M NH_4_Cl + 0.2 M TMAOH, as shown in Figure 3b. Studies demonstrated a clear relationship between the concentration of hydroxide and the etching duration and type of product (amorphous carbon or MXene). Approximately 10 h was needed when the value was 0.2 since the reaction efficiency was great. In addition, calculations using the density-functional theory showed that the etching process took place when Ti_3_AlC_2_ served as the anode and was positively charged. The following conclusions in terms of hypothetical equations were reached in light of the experimental findings [42]:Ti_3_AlC_2_ + 3e^−^ + 3Cl^−^ = Ti_3_C_2_ + AlCl_3_(8)
Ti_3_C_2_ + 2OH^−^ − 2e^−^ → Ti_3_C_2_(OH)_2_(9)
Ti_3_C_2_ + 2H_2_O → Ti_3_C_2_(OH)_2_ + H_2_(10)

## 4. MXenes in LIBs

It is necessary to transform and store the energy harvested from nature, such as that derived from solar and wind sources, into electrochemical energy [46]. The most extensively studied electrochemical energy storage technologies are batteries and super capacitors. To address the growing need for both consumer electronics and electric transportation systems, enormous research efforts are being focused on developing new-generation battery-based electrochemical energy storage technologies [47]. Because of its affordable price, high stability, and remarkable electronic conductivity, graphite is the most frequently utilized anode material for LIBs, which are the primary power source in small portable devices, electric automobiles, and hybrid electric automobiles. However, the specific capacity of the graphite anode is limited to 372 mAhg^−1^ because of the lithium-ion intercalation process [48]. A large number of nanostructures and composites, based on carbon materials, transition metal oxides/hydroxides, and conducting polymers, have been studied with a focus on three key parameters—surface area, conductivity, and pore structure—to enhance the electrochemical performance of battery electrodes [49,50]. Limited specific capacitance, low electrical conductivity, structural deterioration, slowly occurring redox reactions, and restricted ion/electron transport are still among these materials’ primary disadvantages.

MXenes have a wide interlayer spacing, great thermal stability, fast ion and molecule diffusion, an easily adjustable structure, a hydrophilic nature, a high surface area, and large interlayer spacing, unlike the other 2D materials [51]. MXene monolayers are anticipated to be metallic because of their high electron density at the Fermi level (E_F_). The anticipated greater N(E_F_) for MXenes has not been demonstrated to result in higher resistivities than the comparable MAX phases in experiments. The energy locations of the O 2p (~6 eV) and F 2p (~9 eV) bands from the Fermi level of Ti_2_CT_x_ and Ti_3_C_2_T_x_ are both affected by adsorption sites and bond lengths to the termination species [52]. Due to their greater ability for adsorption and reversible intercalation of several metal cations, including Li^+^ ions at the edges and interlayer gaps of MXenes, 2D MXene materials reflect the features of metal-like conductivity and hydrophilic behavior. In addition, MXenes are excellent candidates for energy storage and conversion applications because of their outstanding qualities, which include superior electrical conductivity, faster ion and molecule diffusion, low working voltage, and a substantial theoretical storage capacity. These parameters attract the interest of researchers to investigate the electrochemical performance of several MXene materials as anodes of LIBs, as shown in Figure 4. 

### 4.1. Mono-Transition Metal MXene as Anode of LIBs

A mono-transition metal MXene is one in which the M layer in the M_n+1_X_n_T_x_ formula is composed of only a single type of transition metal (i.e., Sc, Ti, V, Zr, Nb, Mo, etc.), such as Ti_3_C_2_, V_2_C, Ti_2_C, and Mo_2_CT_x_ etc. These types of MXene materials possess superior ionic and electric conductivity, larger specific surface area, and excellent cation intercalation ability, and have demonstrated outstanding potential as anodes of LIBs. Naguib et al. [53] prepared a 2D layered exfoliated Ti_2_C MXene, which was created by etching of Al from Ti_2_AlC. In comparison to untreated Ti_2_AlC, the newly produced Ti_2_C showed a significant (∼10-fold) increase in surface area. In addition, it exhibited a specific capacity of 225 mAhg^−1^, which was significantly greater (almost 5 times) than that of the Ti_2_AlC precursor. After 120 cycles at a 3 C rate and 200 cycles at a 10 C rate, consistent cycling capacities of 80 mAhg^−1^ and 70 mAhg^−1^ were observed, respectively. The greater surface area, open structure, and weaker bonding between MX layers in the Ti_2_C structure were thought to be the causes of the rise in specific capacity. Additionally, the Li^+^ ions can be accommodated in the interlayer voids between the exfoliated Ti_2_C sheets.

Sun et al. [54] reported a Ti_3_C_2_ MXene by the exfoliation of Ti_3_AlC_2_ with 49% HF solution at a temperature of 60 °C for 24 h, and the subsequent intercalation of the exfoliated powder with dimethyl sulfoxide. An increase in lamellar thickness from 30 nm to 100 nm was observed due to the intercalation of exfoliated powder. Owing to the large d-spacing, the intercalated Ti_3_C_2_ MXene showed superior discharge capacity of 264.5 and 118.7 mAhg^−1^ in the first and seventy-fifth cycles, respectively, compared to the capacities of 107.2 mAhg^−1^ and 89.7 mAhg^−1^ for exfoliated Ti_3_C_2_ at a rate of 1 C. The distinction in the performance between ex-Ti_3_C_2_ and In-Ti_3_C_2_ was due to the enlarged d-spacing for Li^+^ storage resulting from intercalation of exfoliated Ti_3_C_2_ by DMSO. However, etching with HF resulted in the −F terminal groups on the surface, which had a negative impact on conductivity and reduced the material’s potential for energy storage and conversion. 

Zhang et al. [55] synthesized the Ti_3_C_2_T_x_ MXene by using a HCl + LiF etchant, instead of a HF solution. Etching and intercalation processes were accomplished in a single step and a defect-free and flexible Ti_3_C_2_T_x_ MXene paper was obtained by HCl + LiF compared to a HF etchant. As-prepared Ti_3_C_2_ MXene was further delaminated by alcohol and the delaminated MXene exhibited excellent cycling performance with specific capacities of 226.3, 137.9, and 102 mAhg^−1^, corresponding to the current densities of 100, 300, and 1000 mAg^−1^, respectively, after 100 cycles.

A fluorine-free Ti_3_C_2_T_x_ MXene was synthesized by alkali treatment (NaOH) in which the surface was dominated with O and OH terminal groups [56]. As-prepared fluorine-free Ti_3_C_2_T_x_ delivered higher capacity and superior performance in comparison to fluorine containing Ti_3_C_2_T_x_ etched using HF. At a current density of 0.5 Ag^−1^, the alkali-etched Ti_3_C_2_T_x_ (mass loading: 1.24 mg cm^−2^) exhibited a capacity of 106.6 mAhg^−1^ after 250 cycles, which was approximately double the capacity of hydrofluoric acid-etched Ti_3_C_2_T_x_ (mass loading: 1.06 mg cm^−2^) due to its bigger c-lattice parameter and functional groups without fluorine termination. Chen et al. [57] synthesized partially etched Ti_3_C_2_ MXene as an anode of an LIB. The precursor Ti_3_AlC_2_ was etched for 5 min, 0.5 h, 1 h, 2 h, 4 h, and 20 h using 40 wt% HF. The reported performance showed that among all the samples, the MXene obtained after 0.5 h etching (denoted 0.5 h-pe Ti_3_C_2_T_x_) provided optimized performance compared to the others. The 0.5 h-pe Ti_3_C_2_T_x_ MXene with 10 wt% conductive additives provided excellent volumetric capacity of 331.6 mAhcm^−3^ and a superior reversible capacity of 160 mAhg^−1^ after 100 cycles at 1 C. In addition, 99% capacity retention was reported after 1000 cycles at the same current density. The distinction in the performance among the samples may be attributed to the formation of a Li-Al alloy between the planes of Ti_3_C_2_, which contributed, at least in part, to raising the capacity of 0.5 h-pe Ti_3_C_2_T_x_ MXene.

Zhao et al. [58] fabricated 3D porous Ti_3_C_2_T_x_ MXene foam via the sulfur template method. The as-developed porous MXene foam provided massive additional sites for Li^+^ storage, as well as channels for electron and ion transfer, as shown in Figure 5. This porous MXene provided an initial capacity of 455.5 mAhg^−1^, with an initial Coulombic efficiency of 65.5% and a capacity of 314.9 mAhg^−1^ after 300 cycles, at 50 mAg^−1^. Moreover, at various current densities of 1, 2, 10, and 15 Ag^−1^, the supplied capacities were 215.6, 187.4, 133.3, and 112.5 mAhg^−1^, respectively. 

Kong et al. [60] reported the electrochemical performance of a Ti_3_C_2_T_x_ MXene obtained by vacuum calcination at various temperatures of 400, 700, and 1000 °C. The observed initial Coulombic efficiencies of these MXenes were 57 and 62% for Ti_3_C_2_T_x_-400 and Ti_3_C_2_T_x_-700, respectively, which were much higher compared to the result of 45% for the Ti_3_C_2_ MXene. The enhancement in performance after calcination may be attributed to the change in the surface chemistry and formation of anatase TiO_2_ on the MXene surface. Moreover, after 100 cycles, at 1 C the supplied capacities of Ti_3_C_2_T_x_, Ti_3_C_2_T_x_-400, Ti_3_C_2_T_x_-700, and Ti_3_C_2_T_x_-1000 electrodes were 87.4, 126.4, 147.4, and 79.5 mAhg^−1^, respectively. The lower capacity of the Ti_3_C_2_T_x_-1000 electrode was thought to be resulted from the structural compactness, which limited the number of ions that could intercalate. 

Meng et al. [61] synthesized a scrolled-type Ti_3_C_2_ MXene by cold quenching in liquid nitrogen and compared the performance of the Ti_3_C_2_ scroll with the Ti_3_C_2_ sheet as an anode of an LIB for different current densities from 100 to 5000 mAg^−1^. The scroll-type MXene exhibited excellent reversible capacities of 226 mAhg^−1^ (1st cycle), 155 mAhg^−1^ (22nd cycle), 136 mAhg^−1^ (32nd cycle), 113 mAhg^−1^ (42nd cycle), and 89 mAhg^−1^ (62nd cycle), whereas Ti_3_C_2_ MXene sheets achieved lower reversible capacities of 199 (1st cycle), 96 (22nd cycle), 68 (32nd cycle), 51 (42nd cycle), and 30 mAhg^−1^ (62nd cycle), corresponding to the current densities of 100, 500, 1000, 2000, and 5000 mAg^−1^ respectively. The enhanced electrochemical performance was reported to be due to the greater contact of the electrolyte, greater interlayer spacing, and shorter diffusion path for Li^+^ in the scroll structure. Furthermore, the Ti_3_C_2_T_x_ scrolls showed outstanding long-term cycling performance by retaining 81.6% of its initial capacity, which was much better than the capacity retention of 63.3% of Ti_3_C_2_T_x_ sheets after 500 cycles at 400 mAg^−1^.

Zhang et al. [62] synthesized pillared SnS/Ti_3_C_2_T_x_ composites decorated with in situ formed TiO_2_ nanoparticles via solvothermal reaction and annealing treatment, as shown in Figure 6. The SnS/Ti_3_C_2_T_x_ composite exhibited high capacity discharge of 866 mAhg^−1^ at a 500 mAg^−1^ current rate with 99% Columbic efficiency, which was better that that of commercial SnS and individual Ti_3_C_2_. The enhance electrochemical performance was ascribed to the pillar effect of Ti_3_C_2_T_x_ MXenes.

Recently, Dai et al. synthesized self-supported and vertically aligned two-dimensional (2D) heterostructures (V-MXene/V_2_O_5_) of rigid Ti_3_C_2_T_X_ MXene and pliable vanadium pentoxide via an ice crystallization-induced strategy as anodes of LIBs [63]. This thick V-MXene/V_2_O_5_ exhibited 472 and 300 mAh g^–1^ at a current rate of 0.2 A g^–1^; the rate performance with 380 and 222 mAh g^–1^, respectively, was retained at 5 A g^–1^ after 800 charge–discharge cycles. The enhanced electrochemical performance was reported to be due to the vertical channels, which facilitated fast electron/ion transport within the entire electrode while the 3D MXene scaffold provided mechanical strength during Li^+^ insertion/de-insertion. Wang et al. synthesized a Fe_3_O_4_@Ti_3_C_2_ MXene hybrid via a simple ultrasonication of Ti_3_C_2_ MXene and Fe_3_O_4_ nanoparticles, for used as an anode of LIBs [64]. One of the compositions of the Fe_3_O_4_@Ti_3_C_2_ hybrid exhibited high reversible capacities of 747.4 mA h g^−1^ at 1 C after 1000 charge–discharge cycles. In addition, this anode material exhibited outstanding volumetric capacity up to 2038 mA h cm^−3^ at 1 C due to the high compact density of the electrode of the prepared hybrid. Tao et al. fabricated mesoporous polydopamine (OMPDA)/Ti_3_C_2_T_x_ via in situ polymerization of dopamine on the surface of Ti3C_2_T_x_ via employing the PS-b-PEO block polymer as a soft template, as shown in Figure 7. This electrode exhibited an average 1000 mAhg^−1^ of discharge capacity with 92% Columbic efficiency at a 50 mAg^−1^ current rate after 200 cycles. The enhanced electrochemical performance was attributed due to the mesopores, which enhanced the overall capacity and reversibility of the reactions with Li^+^ [65]. 

Huan et al. [66] reported the electrostatic self-assembly of 0D-2D SnO_2_ Quantum Dots/Ti_3_C_2_T_x_ MXene hybrids as anodes for LIBs. This electrode exhibited superior lithium storage properties, with a high capacity of 887.4 mAh g^−1^ at a 50 mA g^−1^ current rate and a stable cycle performance of 659.8 mAh g^−1^ at 100 mA g^−1^ after 100 cycles, with a capacity retention of 91%. It is noted that the enhanced electrochemical performance was attributed to the efficient pathways that enabled fast transport of electrons and Li^+^ by the Ti_3_C_2_T_x_ MXene. In addition, this MXene material buffered the volume change of SnO_2_ during Li insertion/de-insertion by confining SnO_2_ QDs between the nanosheets.

Liu et al. [67] explored the performance of V_2_C MXene as an anode of an LIB. The synthesis process was undertaken by immersing the precursor V_2_AlC in a mixture of NaF and HCl for 72 h. The as-prepared MXene delivered an initial discharge and charge capacity of 467 mAhg^−1^ and 291 mAhg^−1^, with a Columbic efficiency of 62.3%, and retained a reversible capacity of 260 mAhg^−1^ after 20 cycles, at a current density of 370 mAg^−1^. In addition, at increasing current densities of 50, 100, 200, 500, and 1000 mAg^−1^, the observed specific capacities for this MXene electrode were 250, 216, 189, 157, and 137 mAhg^−1^, respectively. Zhao et al. [68] devised a 2D Nb_4_C_3_ MXene, which had greater interlayer space for ion accommodation. This MXene as an anode of an LIB showed outstanding initial discharge and charge capacities of 546 mAhg^−1^ and 333 mAhg^−1^, respectively, at a current density of 100 mAg^−1^. Formation of the SEI layer or the irreversible reduction of the active surface groups, such as hydroxyl or fluorine, was the reason for the capacity loss in the first cycle. Even after 100 cycles, it delivered a reversible capacity of 380 mAhg^−1^ at the same current density. Furthermore, this MXene electrode exhibited excellent rate performances with the capacities of 346, 320, 278, 238, 196, 131, and 88 mAhg^−1^ at the rates of 0.1, 0.2, 0.5, 1, 2, 5, and 10 Ag^−1^, respectively, and a discharge capacity of 365 mAhg^−1^ when the current density returned to 0.1 Ag^−1^.

Zhou et al. [69] reported the effect of ball milling on the electrochemical performance of a V_4_C_3_ MXene as an anode of a high-performance LIB. The ball-milled V_4_C_3_ MXene supplied superior discharge and charge capacities of 600.8 and 376.2 mAhg^−1^ in the first cycle and a high reversible capacity of 225 mAhg^−1^ after 300 charge–discharge cycles; these results were much higher than the capacities of 164.1 and 81.2 mAhg^−1^ in the first cycle, and 123.5 mAhg^−1^ after 300 cycles, for non-ball-milled V_4_C_3_ MXene, at a current density of 0.1 Ag^−1^. The distinction in the performance was attributed to the effect of ball milling on MXene powder, which ensured an enlarged specific surface area and interlayer spacing; this not only provided more active space for Li-ion intercalation, but also reduced the diffusion barrier for ion and electron transport. Dong et al. [70] synthesized a Nb_2_CT_x_ MXene by immersing the Nb_2_AlC precursor in Lewis acid molten salt at a temperature of 750 °C for 5 h. The surface termination of the as-prepared Nb_2_CT_x_ MXene dominated with oxygen, which was beneficial for Li-ion storage. This type of MXene delivered a maximum lithium storage capacity of 330 mAhg^−1^ at a current density of 50 mAg^−1^. Moreover, at high rates of 50 C and 100 C, the observed capacities for this MXene were 113 and 80 mAhg^−1^, respectively.

A one-step synthesis method was employed for the preparation of a few-layered Nb_2_CT_x_ MXene, which exhibited enhanced electrochemical performance compared to a multilayered Nb_2_CT_x_ MXene as an anode of an LIB [71]. The few-layered Nb_2_CT_x_ MXene electrode showed outstanding discharge and charge capacities of 746 and 524 mAhg^−1^, respectively, corresponding to 70.2% Columbic efficiency in the first cycle and a capacity of 485 mAhg^−1^ during the second cycle at a current density of 50 mAg^−1^. After 100 charge–discharge cycles, this few-layered MXene delivered a reversible capacity of 354 mAhg^−1^, which was much higher than the value of 165 mAhg^−1^ for the multilayered Nb_2_C MXene, at the same current density. The superior result of the few-layered Nb_2_C MXene anode could be attributed to the presence of a lower content of F termination on the surface and having a 2.7 times larger specific surface area compared to multilayered MXene, which offered more active area and active sites for Li-ion storage. Furthermore, the few-layered electrode displayed an excellent rate performance with specific capacities of 478, 304, 264, and 211 mAhg^−1^ at the current densities of 0.05, 0.25, 0.5, and 1.0 Ag^−1^, respectively. Luo et al. [72] oxidized the partially etched V_2_CT_x_ MXene in the presence of H_2_O_2_ using a hydrothermal reaction. As an anode of LIB, the oxidized V_2_CT_X_ MXene delivered a higher specific capacity of 318 mAhg^−1^ after 100 cycles than V_2_CT_x_ MXene, which achieved a result of 223 mAhg^−1^ at 100 mAg^−1^. In addition, a better rate performance was observed for oxidized V_2_CT_x_ MXene with capacities of 318, 241, and 124 mAhg^−1^ corresponding to the current densities of 50, 200, and 1000 mAg^−1^, respectively. The enhancement in performance after oxidation may be attributed to the change in surface status (presence of abundant O terminated groups) and the formation of VO_2_ on the interface and surface, which functioned as a bridge to connect Li^+^ and oxidized V_2_CT_x_, and resulted in more diffusion channels.

Guo et al. [73] employed the hydrothermal method to synthesize a Mo_2_CT_x_ MXene by etching Ga from a Mo_2_Ga_2_C precursor. The etching process was carried out with five types of etching solutions of LiF + HCl, NaF + HCl, KF + HCl, NH_4_F + HCl, and HF. The Mo_2_CT_x_-K MXene (obtained by etching with KF + HCl) with residual K on the surface delivered a higher performance compared to the other cleaned surface samples of Mo_2_CT_x_-Li, Mo_2_CT_x_-Na, Mo_2_CT_x_-N, and Mo_2_CT_x_-H, as shown in Figure 8 At a current density of 50 mAg^−1^, the Mo_2_CT_x_-K MXene exhibited superior initial discharge and charge capacity of 1069.6 mAhg^−1^ and 893.6 mAhg^−1^, respectively, with an initial Coulombic efficiency of 83.5% and supplied a stable capacity of 151.7 mAhg^−1^ after 100 cycles at a current density of 100 mAg^−1^. The superior performance of the Mo_2_CT_x_-K MXene compared to the other samples was attributed to the existence of residual K on the surface, which bonded with O of Mo_2_CT_x_ and pillared the 2D sheet. This pillared effect provided enough pathways for Li diffusion and space for Li-ion in the 2D Mo_2_CT_x_ sheet. Further, the Mo_2_CT_x_-K MXene was delaminated using tetrabutyl-ammonium hydroxide (TBAOH). The delaminated Mo_2_CT_x_-K MXene with enlarged interlayer spacing and larger active surface delivered a 7.3 times higher capacity of 239 mAhg^−1^, compared to 32.7 mAhg^−1^ of Mo_2_CT_x_-K MXene, at a current density of 800 mAg^−1^.

Zhou et al. [74] reported the electrochemical performance of a delaminated d-Hf_3_C_2_T_x_ MXene as an anode of an LIB and showed that the d-Hf_3_C_2_T_x_ MXene delivered a first discharge capacity of 145 mAhg^−1^, with a Coulombic efficiency of 59% and a capacity of 146 mAhg^−1^ after 200 cycles, at a current density of 200 mAg^−1^. Du et al. [75] synthesized a freeze-dried Ti_3_CNT_x_ MXene as an anode of an LIB. The as-prepared Ti_3_CNT_x_ MXene supplied an excellent initial discharge capacity of ~590 mAhg^−1^ with an initial Coulombic efficiency of 42%, and a stable reversible capacity of 343 mAhg^−1^ after 200 cycles, at a current density of 0.05 Ag^−1^. Moreover, at varied current densities of 0.05, 0.5, 1, and 2 Ag^−1^, the delivered capacities of this MXene were 215, 174, 142, and 107 mAhg^−1^, respectively.

Ti_3_C_2_ and Nb_2_C MXene were fabricated as anodes of LIBs by cold pressing the MXene powders at a pressure of 1 GPa without the use of any binder [76]. The as-fabricated Ti_3_C_2_ and Nb_2_C MXene maintained a capacity of 97 and 128 mAhg^−1^ after 50 cycles, at a current density of 30 mAg^−1^. Naguib et al. [77] compared the electrochemical performance of Nb_2_CT_x_ and V_2_CT_x_ MXenes, obtained via etching of their respective precursor with 50 wt% HF solution. As an anode of an LIB, the Nb_2_CT_x_ and V_2_CT_x_ MXenes exhibited excellent discharge capacities of 422 and 380 mAhg^−1^ in the first cycle, at 1 C, and maintained a reversible capacity of 110 and 125 mAhg^−1^ after 150 cycles, at 10 C. A UV-induced selective etching method was employed to synthesize fluorine free mesoporous Mo_2_C MXene as an anode of an LIB [78]. The as-prepared Mo_2_C MXene exhibited an excellent rate performance with the capacities of 130, 136, 117, 101, 58, 45, 33, and 19 mAhg^−1^ at the current densities of 5, 10, 20, 50, 500, 1000, 2000, and 5000 mAg^−1^, respectively.

To summarize, Table 1 shows the long-cycle performance of various 2D mono-transition metal MXenes as anodes of LIBs. It shows that the performance of similar types of MXene varies depending on the etching solution, etching time, and temperature. The reason for this change may be the different composition and formation of different terminal groups on the surface of the MXene.

### 4.2. Double-Transition Metal MXene (DTM) as Anode of LIBs

The double-transition metal MXene is comprised of two distinct transition metals denoted M′ and M″. Based on the structure, double-transition MXenes are divided into two categories, as shown in Figure 9. The first is ordered MXenes, in which the two transition metals (M′ and M″) are arranged alternatively in a 2D structure. This material can be further classified into two groups: (i) in-plane ordered DTM, denoted by the formula M′_4/3_M″_2/3_XT, such as Mo_4/3_Y_2/3_CT*_x_*; and (ii) out-of-plane ordered DTM, which is defined by the formula of M′_2_M″X_2_T*_x_* or M′_2_M″_2_X_3_T*_x_*, such as Mo_2_TiC_2_T*_x_*. The other DTM is the solid-solution DTM, which is composed of two distinct transition metals distributed randomly in all M layers and is denoted by the formula of (M′, M″)*_n_*_+1_C*_n_*T*_x_*, such as (Ti,Nb)_3_C_2_T*_x_*. These diverse transition metal arrays and structures of DTM resulted in improved thermoelectric, mechanical, and electrochemical properties [21].

Cheng et al. [79] synthesized a novel phased VNbCT_x_ double-transition MXene by selectively etching the Al layer from its VNbAlC MAX phase. As an anode of an LIB, the VNbCT_x_ MXene delivered an excellent specific capacity of 520.5 mAhg^−1^, at a current density of 0.1 Ag^−1^. The capacity was increased gradually with the cycle number, which may be because of the rapid transfer of ions and electrons in the layered nanosheet of the VNbCT_x_ MXene. A long-term cyclic stability (1000 cycles) was also observed in this MXene, even though the structure curled from the nanosheet after a long cycle; this may be attributed to the wrinkled layered structure, which provided channels for Li^+^ transport. In addition, this electrode provided a promising rate performance with specific capacities of 409.7, 301.7, 222.5, 133.5, 81.4, and 53.7 mAhg^−1^ corresponding to the current densities of 0.2, 0.5, 1, 2, 5, and 10 Ag^−1^ respectively. This superior performance resulted from the multilayer skeleton of the VNbCT_x_ MXene, which prevented the restacking of sheets and limited the volume expansion during cycling.

Liu et al. [80] successfully fabricated a highly conductive Ti_2_NbC_2_T_x_ double-transition metal MXene, which delivered a higher electrochemical performance in comparison to the Ti_3_C_2_T_x_ MXene as an anode of an LIB. At a current density of 0.1 Ag^−1^, the Ti_2_NbC_2_T_x_ electrode supplied a discharge capacity of 305.7 mAhg^−1^, corresponding to an efficiency of 58.65% in the first cycle, whereas the delivered capacity of the Ti_3_C_2_T_x_ MXene was 502.2 mAhg^−1^, with an efficiency of 44.2%. The higher efficiency of the Ti_2_NbC_2_T_x_ electrode was attributed to the lower charge transfer resistance and higher extraction/insertion rate of Li^+^. However, after 400 cycles, the displayed capacity of Ti_2_NbC_2_T_x_ was 198 mAhg^−1^, which was higher than the result of 135 mAhg^−1^ for Ti_3_C_2_T_x_, under the same current density. Even at varied current densities of 0.1, 0.2, 0.5, 1.0, 2.0, and 5.0 Ag^−1^, the Ti_2_NbC_2_T_x_ MXene still showed a superior rate performance of 196.2, 156.6, 141.4, 124.8, 113.6, and 90.6 mAhg^−1^, compared to 135.3, 89.8, 57.6, 35.9, 18.8, and 7.8 mAhg^−1^ of the Ti_3_C_2_T_x_ MXene. Moreover, the Ti_2_NbC_2_T_x_ exhibited excellent capacity retention of 81%, after 4000 cycles, under a current density of 1 Ag^−1^. This improved capacity may be attributed to the incorporation of Nb, which enlarged the interplanar spacing as well as providing increased active sites.

A series of (V_x_Ti_1−x_)_2_C (x = 0, 0.3, 0.5, 0.7, 1) MXene compounds were synthesized by etching of (V_x_Ti_1−x_)_2_AlC solid solution for different time periods of 1 h, 5 h, 24 h, 36 h, and 48 h [81]. As an anode of an LIB, the (V_0.5_Ti_0.5_)_2_C-24 h MXene delivered an initial discharge and charge capacity of 445.9 and 286.6 mAhg^−1^, respectively, with 64% Coulombic efficiency and a capacity of 194.9 mAhg^−1^ after 100 cycles at a current density 1 Ag^−1^. Formation of SEI on the surface and irreversible reduction were thought to be the cause of the irreversible capacity in the initial cycles. After 500 cycles, the supplied capacities of Ti_2_C-1 h, (V_0_._3_Ti_0.7_)_2_C-5 h, (V_0_._5_Ti_0.5_)_2_C-24 h, (V_0_._7_Ti_0.3_)_2_C-36 h, and V_2_C-48 h samples were 128.9, 132.5, 194.9, 155.3, and 115.6 mAhg^−1^, respectively, at the same current density. The lower capacity of V_2_C-48 h and Ti_2_C-1 h MXenes was attributed to the restacking of the MXene sheet, and the superior capacity of the (V_0_._5_Ti_0.5_)_2_C-24 h MXene was ascribed to the maximum synergy of Ti and V. Syamsai et al. [82] prepared a layered Ti_x_Ta_(4−x)_C_3_ bi-metal MXene as an anode of an LIB, which initially delivered a discharge capacity of 1411 mAhg^−1^ at a rate of 0.05 C, and retained a reversible capacity of 476 mAhg^−1^, with a Coulombic efficiency of ~99% after 100 cycles at 0.5 C. This outstanding performance was attributed to the formation of the stable bi-metallic MXene with expanded interlayer d-spacing of 3.37 Å, which allowed storage of Li^+^ on its surface and layers. It is noted that (Nb_0.8_, Ti_0.2_)_4_C_3_T_x_ and (Nb_0.8_, Zr_0.2_)_4_C_3_T_x_ MXenes synthesized via altering 20% of Nb from a Nb_4_C_3_T_x_ MXene improved the Li^+^ storage capacity [83]. However, the result was not comparable. It was noted that after 20 cycles, the (Nb_0.8_, Ti_0.2_)_4_C_3_T_x_ and (Nb_0.8_, Zr_0.2_)_4_C_3_T_x_ MXenes showed lower capacity of 158 and 132 mAhg^−1^ compared to the capacity of Nb_4_C_3_T_x_ MXene, which was 189 mAhg^−1^. Several groups experimentally investigated the DTM as an anode of LIBs. Their electrochemical performances are summarized in Table 2.

### 4.3. Composite MXenes as Anodes of LIBs

The mono- and double-transition metal MXenes have good electrical conductivity and cyclic stability. However, various challenges were observed in achieving high specific capacity, high stability, and effective electron/ion transport, since most 2D MXene nanosheets are vulnerable to self-staking, which reduces specific surface area and Li^+^ intercalation numbers. To increase specific surface area and facilitate ion mobility, one strategy is to expand the interlayer gaps, which can be accomplished by applying different types of doping and fillers (Si, Sn, Ag, Fe, CNT, and other oxides) on the surface of the MXene. The combinations of MXenes with other dopants and fillers are defined as composite MXenes, such as Ti_3_C_2_/Si, SiO_2_/Ti_3_C_2_T_x_, and Fe/Ti_3_C_2_T_x_. The typical schematic representation of synthesis process of MXene/Metal composite as shown in Figure 10.

Li et al. [84] fabricated a nickel hydroxide-based delaminated titanium carbide (Ni(OH)_2_/d-Ti_3_C_2_) composite MXene as an anode of a lithium-ion battery. The well-wrapped Ni(OH)_2_ tackled the restacking of d-Ti_3_C_2_ and enlarged the surface area for Li^+^ insertion, as well as shortening the Li^+^ diffusion distance. The as-prepared composite MXene exhibited better performance than delaminated Ti_3_C_2_, and delivered an initial discharge and charge capacity of 615.2 mAhg^−1^ and 578.7 mAhg^−1^ with an efficiency of 94.06%, and a reversible capacity of 732.6 mAhg^−1^, after 200 cycles at a current density of 0.1 Ag^−1^; this was 6 times superior to that of delaminated Ti_3_C_2_ MXene (121.3 mAhg^−1^) at the same current density.

Zhang et al. [85] prepared the NiCo-LDH/Ti_3_C_2_ composite MXene via the electrostatic interaction force between the surface of the negatively charged Ti_3_C_2_ MXene and positively charged nickel-cobalt ions. The tightly anchored ultra-thin, bent, and wrinkled α-phase crystal of NiCo-LDH and the vertical development of LDH in the three-dimensional conductive network of the NiCo-LDH/Ti_3_C_2_ MXene resulted in a gap of 8.1 Å between the interlayers and supplied more active sites that inhibited MXene restacking and accelerated ion diffusion. As a result, the NiCo-LDH/Ti_3_C_2_ MXene delivered superior initial discharge and charge capacities of 1827.3 and 1266.4 mAhg^−1^ respectively, at a current density of 100 mAg^−1^. In addition, at varied current densities from 0.1 to 10 Ag^−1^ the observed discharge capacities were in the range of 1076.7 to 370.6 mAhg^−1^. Li et al. [86] synthesized the NiFe-LDH/Ti_3_C_2_T_x_ MXene via the hydrothermal method and analyzed the performance as an anode of an LIB. The as-prepared NiFe-LDH/Ti_3_C_2_T_x_ MXene delivered a superior initial discharge capacity of 1376.4 mAhg^−1^, with an initial Coulombic efficiency of 56.6%, and exhibited excellent cycling stability with a capacity of 898.8 mAhg^−1^, after 200 cycles, at a current density of 0.1 Ag^−1^. In addition, this electrode supplied high rate performances, with the capacities of 959.5, 651.7, 528.8, 413.8, 315.5, and 270.3 mAhg^−1^ corresponding to the current densities of 0.1, 0.2, 0.5, 1, 1.5 and 2 Ag^−1^, respectively.

Hui et al. [87] prepared a Ti_3_C_2_/Si composite MXene as an anode of a lithium-ion battery. This Ti_3_C_2_/Si composite electrode exhibited very high first discharge/charge capacities of 3512.5/2145.1 mAhg^−1^, with an initial efficiency of 61.1% and an efficiency of 87.8% in the second cycle at a current density of 100 mAg^−1^. In addition, this MXene displayed an excellent cycling stability, with an improved capacity of 1475 mAhg^−1^ after 200 cycles at the same current density. The excellent performance of this MXene could be attributed to the uniform dispersion of Si nanoparticles over the surface of the Ti_3_C_2_ MXene sheet, which ensured faster Li^+^ and electron transportation channels and prevention of volume expansion of Si by Ti_3_C_2_ during cycling. Zhou et al. [88] fabricated a Si-nanosphere-coated Ti_3_C_2_Tx composite MXene (denoted Si/Ti_3_C_2_T_x_) as an anode of an LIB via the electrostatic self-assembly method. The as-assembled MXene exhibited excellent first-cycle discharge and charge capacities of 3986.8 and 3025.1 mAhg^−1^, respectively, and delivered a reversible capacity of 2442.5 mAhg^−1^, after 100 cycles, at a current density of 0.1 Ag^−1^. In addition, a much higher reversible capacity of 1917.9 mAhg^−1^ was observed, compared to the result of 46.9 mAhg^−1^ for pristine Ti_3_C_2_ MXene after 300 cycles at a current density of 0.5 Ag^−1^.

Tian et al. [89] revealed the electrochemical performance of a flexible and binder-free Si/Ti_3_C_2_T_x_ composite MXene as an anode of an LIB. The as-prepared binder-free anode delivered a discharge capacity of ~2930 mAhg^−1^, with a Coulombic efficiency of 71% in the first cycle and a capacity of 2118 mAhg^−1^ after 100 cycles, at a current density of 200 mAg^−1^. Moreover, at varied current densities from 1000, 2000, 3000, and 4000 to 5000 mAg^−1^, the supplied discharge capacities of this MXene were 1768, 1501, 1294, 1033, and 886 mAhg^−1^, respectively. Zhang et al. [90] prepared a flexible porous Si/Ti_3_C_2_T_x_ composite MXene by vacuum filtration and fabricated an anode of an LIB using this composite MXene with various mass ratios of Si and Ti_3_C_2_T_x_ MXene (1:1, 2:1, and 3:1), as shown in Figure 11. The higher mass content of porous Si resulted in the enhancement of electrochemical performance. However, the structure of Si/Ti_3_C_2_T_x_-3:1 was more brittle and fragmented during cell assembly, which led to capacity failure after a few cycles. Hence, the Si/Ti_3_C_2_T_x_-2:1 composition provided optimum performance among all combinations, with initial discharge and charge capacities of 2843.5 and 1778.4 mAhg^−1^, respectively, and maintained a reversible capacity of 1039.3 mAhg^−1^ after 200 cycles at a current density of 500 mAg^−1^. In addition, at current densities of 0.05, 0.5, and 5 A/g, the observed capacities of Si/Ti_3_C_2_T_x_-2:1 MXene were 2256.5, 1661, and 840.3 mAhg^−1^, respectively, which were much higher than those of porous Si and other ratios of Si/Ti_3_C_2_T_x_. 

Bashir et al. [91] embedded silicon nanoparticles on a V_2_C MXene nanosheet to create a Si-V_2_C nanocomposite MXene electrode for an LIB. This composite electrode exhibited excellent first-cycle discharge and charge capacity of 921 and 691 mAhg^−1^, respectively, and retained a stable capacity of 430 mAhg^−1^ after 150 cycles, at a current density of 200 mAg^−1^. In addition, the Si-V_2_C electrode displayed a high-rate performance, with a capacity of 140 mAhg^−1^ corresponding to a current density of 3 Ag^−1^.

Maughan et al. [92] reported the electrochemical performance of a Mo_2_TiC_2_–Si-400 MXene, obtained via amine-assisted silica pillaring and calcination at 400 °C. The pillaring method created a porous Mo_2_TiC_2_ MXene, which resulted in larger interlayer spacing up to 4.2 nm. As a result, the Mo_2_TiC_2_–Si-400 MXene delivered initial discharge and charge capacities of 473 and 314 mAhg^−1^, with an initial Coulombic efficiency of 66% and an efficiency of 94% in the second cycle at a current density of 20 mAg^−1^. Moreover, at the current densities of 20, 50, 200, 500, and 1000 mAg^−1^, the observed discharge capacities of this MXene were 312, 281, 229, 182, and 143 mAhg^−1^, respectively. Mu et al. [93] synthesized a microsphere-like hybrid SiO_2_/Ti_3_C_2_T_x_ MXene and revealed its performance as an anode of an LIB at different current densities. At a current density of 200 mAg^−1^, the delivered charge and discharge capacity of this electrode during the first cycle was 820 and 1173 mAhg^−1^, respectively, and after 100 cycles it maintained a discharge capacity of 798 mAhg^−1^ at the same current density. In addition, an excellent rate performance was also observed with the capacities of 840, 739, 683, 553, and 517 mAhg^−1^ at the current densities of 0.1, 0.5, 1, 2, and 3 Ag^−1^ respectively. This exceptional performance might be attributed to the development of a 3D high-conduction network and the bond between MXene and SiO_2_ nanoparticles, which boosted the structural stability and shortened the Li-ion pathway length.

Han et al. [94] designed a self-integrated Si/Ti_3_C_2_T_x_ MXene bonded with an interfacial nitrogen layer (denoted Si-N-Ti_3_C_2_T_x_) as a high-performance anode for an LIB. The interfacial nitrogen bond boosted adhesion between Si and highly conductive Ti_3_C_2_T_x_ MXene, which resulted in improved rate performance and cycling stability by facilitating ion and electron transport. As a result, the Si-N-Ti_3_C_2_T_x_ MXene electrode supplied superior specific capacities of 2228, 2078, 1818, and 1469 mAhg^−1^, corresponding to the current densities of 0.8, 1.6, 3.2, and 6.4 Ag^−1^, respectively, and maintained excellent cyclic stability with a specific capacity of 1060 mAhg^−1^, after 500 cycles, at a current density of 1.6 Ag^−1^.

Zhang et al. [95] produced a SiO_X_-coated N-doped Ti_3_C_2_T_x_ MXene composite as the anode of an LIB by employing melamine-assisted ball-milling and annealing processes. The prepared composite MXene delivered excellent discharge and charge capacities of 1882.1 and 1141.3 mAhg^−1^, respectively, during the first cycle at 100 mAhg^−1^, and supplied a capacity of 1141.3 mAhg^−1^ after 100 cycles, at a current density of 500 mAhg^−1^. Moreover, at different current densities of 0.5, 1, 3, and 5 Ag^−1^ the delivered discharge capacities of this electrode were 1179.7, 1068.1, 708.9, and 596.4 mAhg^−1^, respectively. In parallel, Choudhury et al. [96] reported the electrochemical performance of a doped Si/graphite/V_2_C MXene composite as an anode of an LIB in a potential range of 0.01–3.0 V. The unique structure of the doped Si/G/V_2_C MXene electrode served to buffer the volume expansion of Si, and doping the MXene produced a greater number of electrons and ion transport channels, which resulted in much higher performance, with the capacity of 3682 mAhg^−1^ compared to the capacity of 2833 mAhg^−1^ for the undoped electrode, at a rate of 1 C. In addition, the doped Si/G/V_2_C exhibited a high reversible capacity of 2439 mAhg^−1^ at a high rate of 10 C, and the capacity was restored to 3491 mAhg^−1^ when the current rate returned to 1 C.

Zhao et al. [97] modified the surface of a Ti_3_C_2_ MXene sheet via the atomic dispersion of Fe and analyzed the electrochemical performance of the Fe-Ti_3_C_2_ MXene as an anode of a lithium-ion battery. The as-prepared Fe-Ti_3_C_2_T_x_ electrode exhibited superior rate performance, with the capacities of 564.9, 400, 260.2, 183.7, and 109.8 mAhg^−1^, compared to the capacities of 77, 68.8, 53, 40.8, and 23.9 mAhg^−1^ of the Ti_3_C_2_ electrode at the current densities of 50, 100, 200, 300 m and 500 mAg^−1^, respectively, under −10 °C. The superior performance of the composite anode could be attributed to the formation of a weaker Fe–O bond with O atoms of the surface functional group in the Fe–Ti_3_C_2_T_x_ sheets. This weaker bond produced unsaturated O atoms, which promoted Li^+^ adsorption and improved the capacity. Huang et al. [98] intercalated Fe ions into the interlayer of a pillared few-layered Ti_3_C_2_ MXene. The few-layered Ti_3_C_2_ MXene prevented a large-scale volume change in the Fe nanocomplex during lithiation and de-lithiation processes, and the expanded interlayer provided more lithium-ion storage space. As a result, the Fe–f–Ti_3_C_2_ MXene supplied excellent an initial discharge and charge capacity of 795 and 470 mAhg^−1^, with an initial efficiency of 59.11% at a current density of 50 mAg^−1^. Moreover, after 150 cycles, the composite MXene maintained a capacity of 535 mAhg^−1^, which was much higher that the result of 142 mAhg^−1^ for f–Ti_3_C_2_ at the current density of 500 mAg^−1^.

Wan et al. [99] successfully fabricated a bi-metal (Fe–Ti) oxide/carbon/Ti_3_C_2_T_x_ MXene electrode for a high-performance LIB. The as-prepared composite electrode displayed superior initial discharge and charge capacities of 1462.2 and 1117.9 mAhg^−1^, respectively, corresponding to an efficiency of 77.52%; in addition, it delivered a discharge capacity of 757 mAhg^−1^ after 800 cycles, at a current density of 3 Ag^−1^. The outstanding lithium storage ability of this composite was due to the remarkable conductivity of carbon and the MXene, and the high capacity of bi-metal (Fe-Ti) oxide, as well as the distinctive 2D layered structure, which allowed multiple open channels for quick electrolyte access, and the internal void space, which mitigated the enormous volumetric expansion during cycling. 

Zhang et al. [100] synthesized the N–Ti_3_C_2_/Fe_2_O_3_ nanocomposite MXene by dispersion of iron oxide on a crumpled nitrogen-doped MXene sheet, and compared its performance with Ti_3_C_2_T_x_/Fe_2_O_3_ and N–Ti_3_C_2_, as an anode of an LIB. Among these, the N–Ti_3_C_2_/Fe_2_O_3_ electrode delivered the best performance, with a high reversible capacity of 688 mAhg^−1^ after 100 cycles, at a current density of 1 Ag^−1^. However, following the same cycles, the Ti_3_C_2_T_x_/Fe_2_O_3_ composites and N–Ti_3_C_2_ only exhibited discharge capacities of 438 mAhg^−1^ and 140 mAhg^−1^, respectively. Furthermore, the N–Ti_3_C_2_/Fe_2_O_3_ electrode showed an excellent rate performance with the capacities of 1065, 993, 806, 672, 477, and 304 mAhg^−1^ at the current densities of 0.1, 0.2, 0.5, 1, 2, and 4 Ag^−1^ respectively. The high performance of the N–Ti_3_C_2_/Fe_2_O_3_ nanocomposite was attributed to the N doping on the MXene sheet, which boosted its overall electronic conductivity and crumpled structure, thereby leading to a larger specific surface area. In addition, the N–Ti_3_C_2_ nanosheets and Fe_2_O_3_ nanoparticles functioned as mutual spacers in the composite to successfully stop the nanoparticles from aggregating and the MXene nanosheets from stacking. He et al. [101] investigated the electrochemical performance of a β-FeOOH nanorod-intercalated β-FeOOH/Ti_3_C_2_T_x_ composite MXene as an anode of a LIB. This sandwich-like β-FeOOH/Ti_3_C_2_T_x_ MXene provided a high discharge capacity of 1630 mAhg^−1^ during the first cycle and retained a capacity of 938 mAhg^−1^ at the end of 100 cycles, at a current density of 200 mAg^−1^. In addition, this MXene displayed a good rate performance with a capacity of 791 mAhg^−1^, at 1 Ag^−1^.

Nam et al. [102] incorporated functionalized titanium carbide nanorods on the surface of a Ti_3_C_2_T_x_ MXene nanosheet and revealed its performance as anode of a LIB. The as-prepared FTCN-Ti_3_C_2_ MXene displayed excellent discharge capacities of 1171 mAhg^−1^, 1077 mAhg^−1^, and 1034 mAhg^−1^ in the 1st, 2nd, and 250th cycles, respectively, at a rate of 0.1 C. Moreover, at varied rates of 0.1, 0.3, 0.5, 1, and 3 C, the supplied capacities were 1133, 1023, 962, 843, and 692 mAhg^−1^, respectively. Lv et al. [103] produced a Ti_3_C_2_ MXene-based carbon-doped TiO_2_/Fe_2_O_3_ composite designated C-TiO_2_/Fe_2_O_3_-Ti_3_C_2_, and analyzed the performance of this MXene as an anode of an LIB, as shown in Figure 12. The C-TiO_2_/Fe_2_O_3_-Ti_3_C_2_ composite anode delivered a better initial discharge capacity of 538 mAhg^−1^, at a current density of 0.1 Ag^−1^. Moreover, at varied current densities of 0.2, 0.5, 1, 2, and 5 Ag^−1^, the supplied discharge capacities for this electrode were 386, 320.5, 274.1, 218.1, and 152.6 mAhg^−1^, respectively. This remarkable electrochemical performance was attributed to the combined influence of carbon doping, layered TiO_2_ structures, and hybridization of Fe_2_O_3_, all of which greatly accelerated the transfer of charges.

Wu et al. [104] explored the electrochemical performance of a Sn-nanoconfined Ti_3_C_2_T_x_ composite MXene as an anode of an LIB. In the Sn–Ti_3_C_2_T_x_ MXene, the Sn nanoparticles enclosed between the spaces of the Ti_3_C_2_T_x_ MXene acted as a pillar, and helped to increase the interlayer space and restricted the sheet stacking, as well as revealing additional sites for Li-ion storage, whereas Ti_3_C_2_ limited the volume expansion of Sn nanoparticles during cycling. Hence, this MXene electrode exhibited good initial discharge and charge capacities of 445 and 374.7 mAhg^−1^, respectively, at a current density of 200 Ag^−1^, and maintained a capacity of 186.9 mAhg^−1^ after 180 cycles, corresponding to a current density of 100 mAg^−1^.

Luo et al. [105] prepared a PVP-Sn(IV)@Ti_3_C_2_ nanocomposite MXene by successful anchoring of Sn^4+^ on the alkalized Ti_3_C_2_ MXene via electrostatic interaction. The presence of PVP assisted in reducing the particles’ size and stopped the material from clumping together during the chemical reaction. In addition, strong chemical adsorption between Sn^4+^ and the negatively charged alkalized Ti_3_C_2_ MXene surface with –OH and –F resulted in superior electrochemical performance of PVP-Sn(IV)@Ti_3_C_2_ nanocomposite MXene compared to graphite and pristine Ti_3_C_2_ MXene. During testing in the range of 0.01 to 3V, the displayed first-cycle discharge and charge capacities of this composite were 1487 and 851 mAhg^−1^, respectively, and a capacity of 635 mAhg^−1^ was observed after 50 cycles at a current density of 0.1 Ag^−1^. Ahmed et al. [106] reported a uniformly coated Ti_3_C_2_ MXene sheet using various thickness of SnO_2_ nanoparticles with the help of an ALD reactor at two different temperatures of 150° and 200 °C. The coating thicknesses of SnO_2_ were varied from 5 to 50 nm. The SnO_2_/Ti_3_C_2_ (ALD@200) MXene with a 50 nm coating thickness supplied superior initial discharge and charge capacities of 1463 and 1041 mAhg^−1^, compared to the capacities of 1024 and 583 mAhg^−1^ of 10 nm coated SnO_2_/Ti_3_C_2_ (ALD@200) MXene, at the current density of 100 mAg^−1^. However, the capacity of 50 nm coated MXene faded drastically due to the inability of the MXene sheet to accommodate the volume change of a large quantity of SnO_2_ nanoparticles, and capacities of 239 (ALD@150) and 451 mAhg^−1^ (ALD@200) were shown after 50 cycles with nearly 50% capacity retention, at 500 mAg^−1^. Furthermore, to overcome the capacity-fading problem, the surface of MXene was coated by HfO_2_ using ALD to form a dual oxide on the surface, which led to improved cyclic stability with the capacity of 843 mAhg^−1^ after 50 cycles, at a 500 mAg^−1^ current rate.

Liu et al. [107] intercalated the SnO_2_ nanoparticles over the surface of a V_2_CT_x_ MXene and reported the performance of the V_2_CT_x_-SnO_2_ MXene as an anode of an LIB for different ratios of V_2_CT_x_ and SnO_2_. At a current density of 50 mAg^−1^, the five samples of V_2_C, such as 1:0.5 V_2_CT_x_-SnO_2_, 1:1 V_2_CT_x_-SnO_2_, 1:2 V_2_CT_x_-SnO_2_, and SnO_2_, delivered the initial discharge/charge capacities of 845/554, 1739.9/1022.03, ~2410.8/~1525.85, ~2449.4/~853.69, and 2728/~1123 mAhg^−1^, respectively, corresponding to the Coulombic efficiencies of 65.5%,~58.7%, ~63.3%, ~34.8%, and ~41.2%, respectively. After 10 cycles, the discharge capacities reached 410, 799.7, 1226.8, 806.2, and 956.5 mAhg^−1^, respectively. Furthermore, after 200 cycles, when the current density reached 1 Ag^−1^, the specific discharge capacities of the five samples were obtained as 143.89, 329, 768, 413.4, and 0.9 mAhg^−1^, respectively. Fan et al. [108] anchored the nanoparticles of Sn_4_P_3_ over the surface of a Ti_3_C_2_T_x_ MXene to produce a Sn_4_P_3_-Ti_3_C_2_T_x_ composite MXene anode for a high-performance LIB. The prepared Sn_4_P_3_-Ti_3_C_2_T_x_ anode showed superior performance to that of the conventional anode, with the initial discharge and charge capacities of 1138 and 936 mAhg^−1^, respectively, at a current density of 0.1 Ag^−1^. Moreover, at different current densities of 0.1, 0.5, 1, and 5 Ag^−1^, the observed discharge capacities of this electrode were 963, 744, 703, and 582 mAhg^−1^, respectively. The high electrochemical and structural stabilities were obtained due to the sandwich structure and potential internal links in the resultant Sn_4_P_3_-Ti_3_C_2_T_x_ hybrid.

Li et al. [109] produced a Ti_3_C_2_ composite MXene decorated by a SnS_2_/Sn_3_S_4_ hybrid through solvothermal and calcination processes. The as-fabricated MXene delivered excellent first-cycle discharge and charge capacities of 1348 mAhg^−1^ and 707.16 mAhg^−1^, respectively, corresponding to a Coulombic efficiency of 37.2%, and retained a specific capacity of 426.3 mAhg^−1^ after 100 cycles at a current density of 100 mAg^−1^. In addition, at increasing current densities of 200, 500, 1000, and 5000 mAg^−1^, the observed capacities for this MXene were 540.4, 479.4, 423.9, and 216.5 mAhg^−1^, respectively. The SnS_2_/Sn_3_S_4_ nanoparticles with a large surface area in the composite acted as a spacer to lessen the propensity of layer stacking and improved the contact between the electrolyte and electrode, which resulted in enhancement of the Li-ion storage capacity of this MXene.

Zhu et al. [110] synthesized a SnO_2_–Ti_2_C–C composite anode by homogeneous coating of graphite with SnO_2_–Ti_2_C nanoparticles. The SnO_2_–Ti_2_C–C anode provided an excellent and long-term cycling stability, as SnO_2_ possessed many active sites and created shorter channels for charge transfer, and graphite limited the volume expansion during cycling. Therefore, this anode exhibited superior initial charge and discharge capacities of 1741.1 and 2167.3 mAhg^−1^, with a Coulombic efficiency of 80.3%. For the second and third cycles, the efficiency reached 95.1% and 98.1%, respectively. At different current densities of 0.2, 0.5, 1, 2, and 3 Ag^−1^, the supplied specific capacities were 1231.32, 998.76, 812.54, 617.63, and 525.36 mAhg^−1^ respectively. When the current density was 0.2 Ag^−1^, the SnO_2_–Ti_2_C–C anode retained a specific capacity of 1036.87 mAhg^−1^ after 200 cycles. Zuo et al. [111] fabricated a nanostructured Sn/SnO_x_-Ti_3_C_2_T_x_ composite MXene electrode by insertion of Sn/SnO_x_ nanoparticles on the surface of a Ti_3_C_2_T_x_ nanosheet through electrostatic attraction and liquid phase reduction. The as-prepared electrode provided a specific capacity of 1169.4 and 1981.3 mAhg^−1^ during the first charge–discharge cycle by retaining a Coulombic efficiency of 59%, whereas 1473.9 and 481 mAhg^−1^ discharge capacities were observed in the case of pure Sn/SnO_x_ and Ti_3_C_2_T_x_, corresponding to the efficiencies of 36.4% and 35.3%, respectively. This superior performance of Sn/SnO_x_-Ti_3_C_2_T_x_ was attributed to the prevention of volume expansion and agglomeration of nanoparticles by Ti_3_C_2_T_x_ and pillaring of Sn/SnO_x_, which overcame sheet stacking and enhanced the Li^+^ storage by broadened the interlayer spacing. 

Tariq et al. [112] synthesized a Ti_3_C_2_/TiO_2_ composite MXene by applying TiO_2_ over the 2 and 5 wt% Ti_3_C_2_ MXene solution. As an anode of a lithium-ion battery, the 5 wt% Ti_3_C_2_/TiO_2_ and 2 wt% Ti_3_C_2_/TiO_2_ MXenes exhibited initial discharge capacities of around 200 mAhg^−1^ and 183 mAhg^−1^, respectively, and supplied discharge capacities of 180 mAhg^−1^ and 165 mAhg^−1^, respectively, after 100 cycles, at 0.1 C. The better result of the 5 wt% Ti_3_C_2_/TiO_2_ MXene was due to the larger surface area of 77.78 m^2^/g compared to the surface areas of 55.68 and 16.25 m^2^/g of 2 wt% Ti_3_C_2_/TiO_2_ MXene and pristine MXene, respectively. Moreover, in 5 wt% Ti_3_C_2_/TiO_2_ MXene, the TiO_2_ fully covered the surface and effectively overcame the layer stacking, whereas in the case of 2 wt% Ti_3_C_2_/TiO_2_, the TiO_2_ agglomerated due to its insufficient area. Jia et al. [113] derived Ti_3_C_2_T_x_ MXene from Ti_3_AlC_2_ by using 46 wt% HF as the etchant and decorated its surface using TiO_2_ nanoparticles to create a Ti_3_C_2_/TiO_2_ composite MXene. As an anode of an LIB, the supplied discharge capacities of this composite MXene for the first three cycles were 552, 301, and 298 mAhg^−1^, respectively, at a scan rate of 0.1 mvs^−1^. The capacity decay was due to the structural damage of the anode and the formation of SEI film on the electrode. At increasing current densities of 0.1, 0.2, 0.5, and 1 Ag^−1^, the observed capacities for the MXene were 296, 275, 221, and 188 mAhg^−1^, respectively. In addition, after 100 cycles, this MXene displayed an excellent capacity of 275 mAhg^−1^ at 0.2 Ag^−1^. Ahmed et al. [114] generated TiO_2_ nanocrystals over the surface of a Ti_2_CT_x_ MXene sheet by oxidation in H_2_O_2_ solution at room temperature. The as-synthesized TiO_2_/Ti_2_C hybrid MXene displayed superior performance as an anode of an LIB. At the current densities of 100, 500, and 1000 Ag^−1^, the supplied discharge capacities of this electrode were 1015, 826, and 681 mAhg^−1^, respectively, in the first cycle, and in the second cycle the discharge capacities were 507, 429, and 384 mAhg^−1^, respectively; moreover, after 50 cycles, the delivered discharge capacities were 389, 337, and 297 mAhg^−1^, respectively. Zhang et al. [115] revealed the performance of a few-layered MoS_2_-wrapped Ti_3_C_2_T_x_ MXene decorated with TiO_2_ nanoparticles (denoted Ti_3_C_2_/TiO_2_@f-MoS_2_) as an anode of an LIB. The multilayered Ti_3_C_2_ with TiO_2_ nanoparticles effectively improved the stability by providing abundant interspace for electron transport, preventing the aggregation of f-MoS_2_ and offering sufficient space. In contrast, the f-MoS_2_ with a larger interlayer facilitated lithium-ion diffusion and restrained the restacking of multilayer Ti_3_C_2_. As a result, the Ti_3_C_2_/TiO_2_@f-MoS_2_ MXene delivered excellent first-cycle discharge and charge capacities of 910.7 and 685.2 mAhg^−1^, with a high Coulombic efficiency of 75.2% and a capacity of 490.7 mAhg^−1^ after 100 cycles, at a current density of 100 mAg^−1^. In addition, at increasing current densities from 0.1 to 5 Ag^−1^, the supplied capacities varied from 613.1 to 40.3 mAhg^−1^.

He et al. [116] modified the surface of a Ti_3_C_2_T_x_ MXene by LaF_3_ and compared its performance with a pristine Ti_3_C_2_T_x_ MXene as an anode of an LIB. It was observed that the Ti_3_C_2_T_x_-LaF_3_ electrode supplied higher performance with initial discharge and charge capacities of 340 and 223 mAhg^−1^, respectively, in comparison to the discharge and charge capacities of 238 and 131 mAhg^−1^ for pristine Ti_3_C_2_T_x_ at a current density of 50 mAhg^−1^. Moreover, at a current density of 1000 mAg^−1^, the Ti_3_C_2_T_x_-LaF_3_ showed a higher reversible capacity of 89.2 mAhg^−1^, compared to 76.9 mAhg^−1^ of Ti_3_C_2_T_x_, after 50 cycles. The capacity and rate ability of the composite MXene were improved by the formation of LaF_3_ on the surface of the Ti_3_C_2_T_x_ MXene, which lowered the resistance and impedance and enhanced the Li-ion diffusion rate.

Yuan et al. [117] synthesized a sulfur-decorated Ti_3_C_2_ MXene as an anode of an LIB. Due to the incorporation of S, the pore sizes were enlarged from 10 to 22.4 nm and the surface area increased to 91.7 from 64.1 m^2^/g. The S-decorated Ti_3_C_2_ MXene provided an initial discharge capacity of 305 mAhg^−1^ and, after 100, 200, and 400 cycles, the capacities remained at 167.8, 170.2, and 166.3 mAhg^−1^, with high Coulombic efficiencies of 99.76%, 99.92%, and 100%, at a current density of 0.5 Ag^−1^. In addition, regarding the rate performance at varied current densities of 0.5, 1, 1.5, 2, and 2.5 Ag^−1^, the S–Ti_3_C_2_T_x_ MXene delivered much higher discharge capacities of 220.2, 138.5, 126.9, 121, and 117.5 mAhg^−1^, compared to 210.4, 74.7, 60.2, 50.6, and 46 mAhg^−1^, respectively, for the Ti_3_C_2_T_x_ electrode. Zhang et al. [118] recently explored the effect of PVDF and CMC binder on the performance of the S-Ti_3_C_2_T_x_ MXene electrode, as shown in Figure 13. The test was carried out for the S-Ti_3_C_2_T_x_ MXene electrode (mass ratio of 1:7) with CMC and PVDF binder. The electrode with CMC and PVDF binder had initial discharge capacities of 1226.8 and 858.8 mAhg^−1^, at a current density of 50 mAg^−1^. As the current density increased to 1000 mAg^−1^, the displayed initial capacities were 372.9 and 117.8 mAhg^−1^; subsequently, after 2550 cycles, the maintained capacities were 858.9 and 424.1 mAhg^−1^ respectively, for CMC and PVDF binder electrodes. Higher adhesion force, better diffusivity of lithium ions, and lowered charge transfer resistance were considered to be the causes of the CMC binder’s superior performance over the PVDF binder.

Zou et al. [119] revealed that the electrochemical performance of a Ti_3_C_2_ MXene improved well with the insertion of Ag nanoparticles on the surface of the MXene sheet. The as-prepared Ti_3_C_2_/Ag MXene displayed superior discharge capacities of 550 and 330 mAhg^−1^ in the first and second cycles, respectively, at a rate of 1 C. On the contrary, the delivered capacities of the Ti_3_C_2_ MXene were 420 and 250 mAhg^−1^ corresponding to the same cycle. In addition, at different current rates of 5, 10, 20, and 50 Ag^−1^, the observed capacities for the Ti_3_C_2_/Ag MXene were 131.5, 100.7, 86.6, and 47.4 mAhg^−1^, respectively. More remarkably, a capacity of 150 mAhg^−1^ was retained even after 5000 cycles, at a rate of 50 C. Wang et al. [120] constructed a Ag-nanoparticle-decorated 3D honeycomb-like hollow Ti_3_C_2_T_x_ composite MXene structure as the anode of an LIB to boost the Li^+^ storage performance compared to a 2D Ti_3_C_2_ MXene alone. Ag nanoparticles were uniformly grafted into the 3D hollow multiporous Ti_3_C_2_ MXene framework, which prevented Ti_3_C_2_ MXene flakes from building up or accumulating in layers. The fabricated 3D Ti_3_C_2_T_x_/Ag composite MXene anode provided a high initial capacity of 611.3 mAhg^−1^, corresponding to an efficiency of 57.9% at a current density of 100 mAg^−1^. Moreover, at gradually increasing current densities of 0.1, 0.5, 1, 2, 3, and 4 Ag^−1^, the delivered reversible capacities were 680.5, 495.9, 399.9, 349.2, 311.4, and 226.7 mAhg^−1^, respectively.

Huang et al. [121] tested the electrochemical performance of a Li_3_VO_4_/Ti_3_C_2_ composite MXene as an anode of an LIB. The Li_3_VO_4_/Ti_3_C_2_ composite MXene was produced by uniform insertion of Li_3_VO_4_ onto a multilayer Ti_3_C_2_T_x_ MXene using the sol–gel method. This type of anode material provided a superior rate performance, with the capacities of 420, 311, 272, 165, and 117 mAhg^−1^ corresponding to the rates of 0.1, 0.5, 1, 5, and 10 C, respectively. In addition, this composite MXene exhibited much higher cyclic stability of 146 mAhg^−1^, compared to 40 mAhg^−1^ and 71 mAhg^−1^ of LVO and graphite, respectively, after 1000 cycles, at 5 C.

Liu et al. [122] introduced nitrogen onto the surface of the MXene nanosheet by hydrothermal reaction of a Nb_2_CT_x_ MXene with urea. The as-prepared N-Nb_2_CT_x_ MXene provided a surface area of 120 m^2^ g^−1^, which was enlarged by more than two times compared to the 55m^2^ g^−1^ surface area of Nb_2_CT_x_. As an anode of an LIB, the N-Nb_2_CT_x_ MXene delivered a first-cycle capacity of 380 mAhg^−1^, with a Coulombic efficiency of 98%, and retained a reversible capacity of 360 mAhg^−1^ after 100 cycles at a current density of 0.2C. Moreover, after 1500 cycles, the N-doped MXene maintained a capacity of 288 mAhg^−1^, which was much higher than the value of 124 mAhg^−1^ for pristine Nb_2_C, at 0.5 C. Zhong et al. [123] homogeneously anchored MgH_2_ nanoparticles on the surface of Ti_3_C_2_ MXene sheets using a bottom-up self-assembly strategy, and analyzed the performances of the as-prepared composite MXene as anode of an LIB with different weight ratios of MgH_2_ and Ti_3_C_2_. The reported initial capacities of the four types of MgH_2_/Ti_3_C_2_ electrode, where the content of MgH_2_ was 20, 40, 60, and 80%, were 629.5, 792.2, 1051.2, and 830.4 mAhg^−1^, respectively, at a current density of 100 mAg^−1^. The performances were enriched gradually with MgH_2_ content. However, when MgH_2_ content exceeded 60%, the performance started to diminish, as these particles were accumulated on the surface of the MXene. Hence, the MgH_2_-60/Ti_3_C_2_ provided the optimum electrochemical performance among all electrodes and maintained a capacity of 389.3 mAhg^−1^ after 100 cycles, at the same current density. The improved capacity, cyclability, and rate performance were the result of the unique two-dimensional nanoarchitecture; this accelerated the transfer of electrons and lithium ions, and inhibited the pulverization of active materials and, more crucially, the F-Mg bonding between MgH_2_ and Ti_3_C_2_, which prevented the shedding of MgH_2_ nanoparticles into the electrolyte during cycling.

Liu et al. [124] fabricated a GeO_x_/Ti_3_C_2_ composite MXene anode with two different combinations of binder and solvent. The first was a PVDF binder with an NMP-solvent-based anode named GeO_x_/Ti_3_C_2_/PVDF(NMP), and the second was a Li-PAA binder with a DI-water-solvent-based anode named GeO_x_/Ti_3_C_2_/Li-PAA(DI-water). Of these two, the GeO_x_/Ti_3_C_2_/Li-PAA(DI-water) electrode displayed better electrochemical stability, with a capacity of 1026 mAhg^−1^ after 50 cycles, at a current density of 200 mAg^−1^. On the contrary, GeO_x_/Ti_3_C_2_/PVDF(NMP) retained a capacity of 826 mAhg^−1^ after the same number of cycles and the same current density. Melchoir et al. [125] investigated the performance of a Ti_2_CT_x_/electrolytic manganese dioxide (EMD) composite MXene with different weight ratios of Ti_2_CT_x_: EMD (i.e., Ti_2_CT_x_:EMD = 20:80; 50:50; 80:20) as an anode of a lithium-ion battery. The inclusion of EMD with the MXene matrix led to the opening of interlayer gaps and enabled better accessibility of the Li ions. However, the properties were inhibited above a certain ratio of EMD content. Hence, among all compositions, the electrode with a MX:EMD = 80:20 ratio exhibited a lower surface film and charge transfer resistance, and delivered superior performance, with the capacities of 570 mAhg^−1^ in the first cycle, 272 mAhg^−1^ in the second cycle, and 460 mAhg^−1^ after 200 cycles, at the current density of 100 mAg^−1^. In addition, at the current density of 1000 mAg^−1^, a much higher capacity of 160 mAhg^−1^ was observed compared to the values of 90 mAhg^−1^ for Ti_2_CT_x_: EMD (20:80) and 86 mAhg^−1^ with a 50:50 ratio, after 1000 cycles with nearly 100% capacity retention.

Zhang et al. [126] synthesized the heterostructured Bi_2_MoO_6_/Ti_3_C_2_T_x_ MXene by combining positively charged Bi_2_MoO_6_ nanoplates with negatively charged Ti_3_C_2_T_x_ MXene sheets. The as-prepared Bi_2_MoO_6_/Ti_3_C_2_T_x_-30% (Bi_2_MoO_6_:Ti_3_C_2_T_x_ = 70:30) composite MXene exhibited excellent initial discharge and charge capacities of 844.2 and 615.5 mAhg^−1^, respectively, with an efficiency of 72.9% at 100 mAg^−1^. Although the initial Coulombic efficiency of Bi_2_MoO_6_/Ti_3_C_2_T_x_-30% was lower than that of Bi_2_MoO_6_/Ti_3_C_2_T_x_-10% (76.6%) and Bi_2_MoO_6_ (82.6%), the supplied capacity after 200 cycles for Bi_2_MoO_6_/Ti_3_C_2_T_x_-30% (692 mAhg^−1^) was much higher than that of Bi_2_MoO_6_/Ti_3_C_2_T_x_-50% (617.5 mAhg^−1^), Bi_2_MoO_6_/Ti_3_C_2_T_x_-10% (497.6 mAhg^−1^), and Bi_2_MoO_6_ (416.1 mAhg^−1^) at the same current density.

Abdah et al. [127] investigated the performance of an activated carbon (AC) integrated Ti_3_C_2_T_x_ hybrid electrode as an anode of an LIB. This composite electrode displayed an excellent initial discharge capacity of 1194.2 mAhg^−1^ and offered a reversible capacity of 841.8 mAhg^−1^ with capacity retention of 70.5%, after 80 cycles, at a current density of 0.2 Ag^−1^. The superior performance of the AC-Ti_3_C_2_T_x_ electrode could be attributed to the presence of the few-layered unique morphology of the Ti_3_C_2_ MXene; this provided abundant active sites for Li^+^ intercalation, enhanced transportation and the diffusion rate of ions and electrons, and enhanced the inclusion of AC, which acted as an interlayer spacer between the layers of MXene and provided large numbers of ion/electron transfer channels. Liu et al. [128] successfully assembled a VO_2_-NTs/Ti_3_C_2_ MXene as an anode of an LIB. This VO_2_-nanotube-linked Ti_3_C_2_ MXene electrode exhibited superior initial discharge and charge capacities of 2132 and 1164 mAhg^−1^, respectively, with an initial Coulombic efficiency of 54% at 0.1 Ag^−1^. However, Coulombic efficiency was increased to 95% in the next cycle. Moreover, an excellent rate capability was also observed, with a specific capacity of 703 mAhg^−1^ at a current density of 10 Ag^−1^. This may be attributed to the combination of 1D VO2-NT nanostructures with 2D Ti3C2 nanosheets, which created several multidimensional channels for the transfer of ions and electrons, and provided a large number of reaction sites for electrochemical processes.

Ma et al. [129] generated niobium-doped TiO_2_ arrays on the surface of a double-transition TiNbCT_x_ MXene (denoted TiNbC@NTO), and explored the effect of oxidation temperature on the electrochemical performance of this composite as an anode of an LIB. The oxidation process was carried out for three different temperatures, namely, 300, 500, and 700 °C, and the observed performance showed that, at 0.1 Ag^−1^, the supplied charging capacities after 100 cycles for TiNbC@NTO-300, TiNbC@NTO-500, and TiNbC@NTO-700 electrodes were 278.4, 295, and 303.1 mAhg^−1^, respectively. However, during the long-cycle performance test, the TiNbC@NTO-500 electrode exhibited higher stability than the other electrodes, with a capacity of 261 mAhg^−1^ after 500 cycles, at a current density of 1 Ag^−1^. The ideal synergistic impact between the MXene layer and Nb-doped TiO_2_ was thought to be a contributor to this cycling stability.

Xu et al. [130] constructed VNbO_5_ metallic oxide on the surface of a VNbCT_x_ MXene via partial oxidation at 500 °C (denoted as VNbC@VNO-500). Partial oxidations strengthened the bond between VNbO_5_ and the VNbCT_x_ MXene, and effectively improved the electrochemical performance. As an anode of an LIB, the prepared VNbC@VNO-500 MXene provided excellent cyclic stability, with a specific capacity of 400.3 mAhg^−1^ after 100 cycles at a current density of 0.1 Ag^−1^, as shown in Figure 14. In addition, this electrode material displayed promising rate performance with specific capacities of 327.3, 306.6, 270.7, 222.4, 181.7, and 141.2 mAhg^−1^ at the current densities of 0.1, 0.2, 0.5, 1.0, 2.0, and 5.0 Ag^−1^, respectively.

Luo et al. [131] fabricated a sandwich-like structured Na_2_Ti_3_O_7_/Ti_3_C_2_ composite MXene as an anode of an LIB. This composite electrode displayed high discharge and charge capacities of 405 mAhg^−1^ and 337 mAhg^−1^ in the first cycle, and retained a specific capacity of 263 mAhg^−1^, after 100 cycles, at a current density of 0.1 Ag^−1^. Moreover, at various current rates of 0.1, 0.2, 0.4, 0.6, 0.8, 1, and 2 Ag^−1^, the observed capacities for the electrode were reported as 361.7, 288.7, 240.4, 223.6, 209.4, 207.7, and 177.8 mAhg^−1^, respectively.

Gong et al. [132] reported the performance of a TiNb_2_O_7_/Ti_3_C_2_ composite MXene as an anode of an LIB. In the TiNb_2_O_7_/Ti_3_C_2_ structure, the TiNb_2_O_7_ nanoparticles connected with MXene nanoflakes and created a 3D conductive network, which facilitated the transfer of ions/electrons. In addition, this composite MXene offered four times higher conductivity than TiNb_2_O_7_. As a consequence, the TiNb_2_O_7_/Ti_3_C_2_ MXene exhibited improved discharge/charge capacities of 388.4/349 mAhg^−1^ during the first cycle, at a current density of 0.1 C. In addition, at gradually varied rates of 0.1, 0.5, 1, 2, 5, 10, and 20 C, the supplied capacities of this electrode were 346.4, 302.4, 284.2, 262.2, 262.2, 198, and 166 mAhg^−1^, respectively. Qi et al. [133] fabricated a chlorophyll (zinc methyl 3-devinyl-3-hydroxymethyl-pyropheophorbide (Chl)) intercalated Chl@Nb_2_C composite MXene, and compared the performance of three different samples of 1%(wt/wt) Chl@Nb_2_C (mass ratio of Chl:Nb_2_C = 1:100), 2%(wt/wt) Chl@Nb_2_C, and Nb_2_C as the anode materials of an LIB. It was observed that the 1%(wt/wt) Chl@Nb_2_C MXene delivered a higher specific capacity of 384 mAhg^−1^, compared to the results of the other two samples of 363 mAhg^−1^ (2%(wt/wt) Chl@Nb_2_C) and 295 mAhg^−1^ (Nb_2_C), respectively, at a current density of 100 mAg^−1^. In addition, at increasing current densities from 0.1 to 1 Ag^−1^, the observed specific capacities varied from 247 to 126 mAhg^−1^, 215 to 116, and 201 to 102 mAhg^−1^ for 1%(wt/wt) Chl@Nb_2_C, 2%(wt/wt) Chl@Nb_2_C, and Nb_2_C, respectively.

Tian et al. [134] reported the electrochemical performance of a Ti_3_C_2_/CoS_2_ composite MXene as an anode of an LIB. This composite MXene electrode provided superior specific capacity of 405.8 mAhg^−1^ after 100 cycles, at a current density of 0.1 Ag^−1^, and supplied an excellent rate performance, with the capacities of 436.9, 387.2, 312.5, 264.8, 209.5, 179.8, and 133.1 mAhg^−1^ corresponding to the rates of 0.1, 0.2, 0.5, 1, 2, 3, and 5 Ag^−1^, respectively; when the current density switched back to 0.1 Ag^−1^, the displayed capacity was still 390.8 mAhg^−1^. In addition, 100% capacity retention was observed after 1000 cycles at a current density of 1 Ag^−1^. This improved performance was attributed to the formation of the Ti-O-Co bond at the MXene interface, which enhanced the redox kinetics by promoting ionic diffusivity and electric conductivity. Shen et al. [135] synthesized the MoS_2_/Ti_3_C_2_ composite MXene via the solid-state sintering method, and revealed that the MoS_2_/Ti_3_C_2_-10 (the mass ratio of ammonium tetrathiomolybdate was 10) electrode delivered a better electrochemical performance than the Ti_3_C_2_, MoS_2_, or MoS_2_/Ti_3_C_2_-5 electrodes. The observed discharge/charge capacities of these four electrodes were 178/95.4, 1012.7/843.6, 268.6/141.2, and 386.4/296.1 mAhg^−1^ in the first cycle, at 50 mAg^−1^, and the discharge capacities were 63, 3.6, 76.2, and 131.6 mAhg^−1^ for Ti_3_C_2_, MoS_2_, MoS_2_/Ti_3_C_2_-5, and MoS_2_/Ti_3_C_2_-10, respectively, after 200 cycles at a current density of 1000 mAg^−1^.

Chen et al. [136] produced a MoS_2_/Mo_2_TiC_2_T_x_ composite MXene as an anode of an LIB. The prepared composite MXene supplied superior initial discharge and charge capacities of 646 and 554 mAhg^−1^, which were 2.4 and 4.1 times higher than those of a pure Mo_2_TiC_2_T_x_ MXene (268 and 134 mAhg^−1^, respectively), at a current density of 100 mAg^−1^. Compared to the restacked pure Mo_2_TiC_2_T_x_, the MoS_2_/Mo_2_TiC_2_T_x_ had an open structure, which minimized the ion diffusion resistance and confirmed the enhancement in capacity. Moreover, at gradually increased current densities of 100, 200, 500, 1000, 2000, and 5000 mAg^−1^, the obtained capacities for this composite electrode were 523, 484, 407, 315, 182, and 90 mAhg^−1^, respectively. Kamat et al. [137] employed a selenium-enriched and over-oxidized Mo_3_Se_4_-anchored Ti_3_C_2_T_x_ composite MXene as an anode of an LIB. The Mo_3_Se_4_-Ti_3_C_2_T_x_ MXene electrode provided an excellent initial discharge capacity of 1930.32 mAhg^−1^ at a cycling rate of 0.1 C; for further cycles at the same rate, high stable discharge capacities of 1250.76, 1107.25, and 1067.94 mAhg^−1^ were obtained with a Coulombic efficiency of more than 100%. At multiple operating rates of 0.2 C, 0.4 C, 0.6 C, and 1 C, the observed charge capacities were 939.31, 876.72, 857.63, and 809.54 mAhg^−1^, with discharge capacities of 978.24, 904.2, 883.97, and 822.52 mAhg^−1^, respectively. Bai et al. [138] reported the electrochemical performance of a thin-carbon-coated and MoS_2_-anchored V_4_C_3_ MXene (denoted V_4_C_3_ /MoS_2_/C) as an anode of an LIB. The thin carbon coating stabilized the connection between the V_4_C_3_ MXene and MoS_2_ nanosheets, which effectively increased the specific area and shortened the ion diffusion distance. As a result, the V_4_C_3_ /MoS_2_/C electrode provided a significantly improved capacity of 622.6 mAhg^−1^, compared to the capacities of 259.6, 30, and 106.9 for MoS_2_/C, V_4_C_3_, and V_4_C_3_ /MoS_2_/C, respectively, after 450 cycles at a current density of 1 Ag^−1^. In addition, high reversible capacities of 827.0, 778.6, 730.7, 674.0, 620.8, 532.6, and 500 mAhg^−1^ were observed for the V_4_C_3_ /MoS_2_/C electrode at the rates of 0.1, 0.2, 0.5, 1, 2, 5, and 10 Ag^−1^, respectively.

Wei et al. [139] explored the performance of a 3 °C GaInSnZn (the calculated mass percentages of Ga, In, Sn, and Zn were 69.87%, 15.52%, 13.46%, and 1.15%, respectively) liquid-metal-incorporated Ti_3_C_2_T_x_ MXene as a flexible anode of an LIB. Liquid metal was confined within the MXene matrix, which possessed good wettability and excellent electrical conductivity, resulting in an improved battery energy density. This liquid metal MXene electrode delivered excellent discharge capacities of 638.79, 535.89, and 531.61 mAhg^−1^ in the first three cycles respectively, at 20 mAg^−1^. Moreover, at various rates of 50, 100, 200, 500, and 1000 mAg^−1^, the supplied capacities were 507.42, 483.33, 480.22, 452.30, and 404.47 mAhg^−1^, respectively. Zhang et al. [140] depicted the electrochemical performance of a carbon-coated 3D tremella-like Ti_3_C_2_T_x_ (denoted T-Ti_3_C_2_T_x_@C) MXene structure as an anode of an LIB. This T-Ti_3_C_2_T_x_@C MXene electrode exhibited a superior initial charge capacity of 580.9 mAhg^−1^, with an initial Coulombic efficiency of 56.9%, and retained a high discharge capacity of 499.4 mAhg^−1^ after 200 cycles, at 0.2 C. At various rates of 0.2, 1, 5, 10, 20, and 50 C, the delivered capacities were 478, 329.5, 237.9, 194.7, 142.2, and 102.2 mAhg^−1^, respectively.

Wang et al. [141] reported that the interlayer distance of a V_2_C MXene expanded from 0.735 nm to 0.952 nm due to inclusion of Co nanoparticles, which created a strong V–O–Co bonding. The surface area was also increased from 11.6 to 27.3 m^2^ g^−1^. As a result, the cobalt-intercalated V_2_C@Co MXene delivered a superior electrochemical performance, with initial discharge and charge capacities of 1117.3 and 1006.5 mAhg^−1^, compared to the discharge/charge capacities of 686.7/620.7 mAhg^−1^ for a pristine V_2_C electrode, at 0.1 Ag^−1^. In addition, this composite electrode displayed an excellent rate performance, with a capacity of 199.9 mAhg^−1^ at a current density of 20 Ag^−1^. Li et al. [142] synthesized the CoO/Ti_3_C_2_T_x_ composite MXene as an anode of an LIB. This CoO-coated Ti_3_C_2_ MXene had a surface area that was 10 times larger than that of Ti_3_C_2_ MXene, and showed superior initial discharge and charge capacities of 1389 and 850 mAhg^−1^, compared to the value of 415 mAhg^−1^ for the pristine Ti_3_C_2_ MXene, at a current density of 100 mAg^−1^. The small size of CoO provided an easy-to-access pathway for electrolyte diffusion and Li^+^ intercalation by widening the layer gap of Ti_3_C_2_. Moreover, Ti_3_C_2_ prevented the electrode material from crumbling or breaking by acting as a mechanical buffer against the large volume shift. The combined effect of CoO and Ti_3_C_2_ resulted in improved capacity of the CoO/Ti_3_C_2_ composite compared to pristine Ti_3_C_2_.

Zhao et al. [143] incorporated Co_3_O_4_ nanoparticles with a highly conductive Ti_3_C_2_ MXene nanosheet at various ratios of Co_3_O_4_ and Ti_3_C_2_. The higher content of Co_3_O_4_ nanoparticles resulted in lowering the conductivity, as well as increasing the probability of agglomeration of nanoparticles on the MXene sheet and lowering the content of Co_3_O_4_ nanoparticles, leading to restacking of the MXene sheet. Hence, the Co_3_O_4_/Ti_3_C_2_ (1:1 ratio) MXene exhibited the optimum performance, with initial discharge and charge capacities of 2082 and 1326 mAhg^−1^, respectively, and maintained a reversible capacity of 1005 mAhg^−1^ after 300 cycles, corresponding to a current density of 1 C. However, the Co_3_O_4_/Ti_3_C_2_ (1:2, 2:1, and 4:1) MXene delivered 678, ~800, and ~600 mAhg^−1^ reversible capacities after 300 cycles at the same current density of 1 C. Oh and Park [144] produced nitrogen-doped graphitic C-coated Co_3_O_4_ nanocrystals on a Ti_3_C_2_T_x_ MXene nanosheet (termed Co_3_O_4_@NGC/MX) as an anode for an LIB. The Co_3_O_4_@NGC/MX electrode delivered an initial discharge capacity of 1086 mAhg^−1^, with an initial Coulombic efficiency of 67.6%, and an outstanding cyclic stability, with a discharge capacity of 830 mAhg^−1^ after 500 cycles at 1 Ag^−1^. In addition, this electrode provided a superior rate performance, with a discharge capacity of 327 mAhg^−1^ at a current density of 50 mAg^−1^.

Zhang et al. [145] constructed a Cu_2_O/Ti_2_C composite MXene as a high-performance anode of an LIB. The hollow Cu_2_O nanoparticles on the surface of the conductive Ti_2_C MXene created a 3D conductive network, which resulted in superior electrochemical performance, with discharge capacities of 790 mAhg^−1^ and 369 mAhg^−1^, corresponding to the Coulombic efficiencies of 46% and 102% in the first and second cycles, at a current density of 10 mAg^−1^. In addition, this MXene provided high rate performances, with the capacities of 369, 260, 203, and 145 mAhg^−1^ at the current densities of 10, 100, 500, and 1000 mAhg^−1^, respectively.

Lin et al. [146] generated a carbon nanofiber bridge on the surface of a Ti_3_C_2_ MXene sheet (denoted Ti_3_C_2_/CNF). With the aid of CNF, conductive bridges were created between the top and lower Ti_3_C_2_ flakes, which provided efficient channels for quick electron transport between the flakes and reduced contact resistance between the Ti_3_C_2_ particles. As an anode of an LIB, this Ti_3_C_2_/CNF MXene delivered superior initial discharge and charge capacities of 848 mAhg^−1^ and 407 mAhg^−1^, respectively, and a discharge capacity of 320 mAhg^−1^ after 295 cycles, at a rate of 1 C. In addition, at gradually varied rates of 1, 3.5, 8.5, and 30 C, the displayed capacities of this MXene were 320, 180, 145, and 106 mAhg^−1^, respectively. Mashtalir et al. [147] showed that incorporating carbon nanotubes (CNTs) into the MXene structure raised the specific capacities and rate performances by enhancing ion accessibility to the MXene layers. As an anode of an LIB, the CNT-incorporated Nb_2_C MXene electrode supplied excellent initial discharge capacities of 1060, 780, 270, and 160 mAhg^−1^ at the rates of 0.1, 0.5, 10, and 20 C, respectively. Furthermore, a capacity of 370 mAhg^−1^ was observed after 100 cycles, at a rate of 2.5 C. Halim et al. [148] investigated the electrochemical performance of a d-Mo_2_C/CNT MXene as an anode of an LIB, and showed that the d-Mo_2_C/CNT MXene electrode supplied a superior initial discharge capacity of 821 mAhg^−1^, with an initial Coulombic efficiency of 76% at a current density of 10 mAg^−1^, and an excellent rate capability, with a capacity of 250 mAhg^−1^ after 1000 cycles at 5 Ag^−1^.

Ren et al. [14] fabricated a p-Ti_3_C_2_T_x_/CNT MXene via the addition of carbon nanotubes with porous Ti_3_C_2_T_x_. The combination of CNTs and the pores of the MXene resulted in the enhancement of capacity. As an anode of an LIB, the p-Ti_3_C_2_T_x_/CNT MXene supplied a discharge capacity of 790 mAhg^−1^ in the first cycle, and a capacity of ~500 mAhg^−1^ after 100 cycles at a rate of 0.5 C. Moreover, at various rates of 0.1, 10, and 60 C, the delivered capacities were 650, 230, and 110 mAhg^−1^, respectively. Cheng et al. [149] investigated the performance of a vanadium-incorporated Ti_3_C_2_T_x_ MXene as an anode of an LIB. It was observed that the V_0.2_-Ti_3_C_2_T_x_ electrode delivered promising specific capacities of 251.3, 212.0, 183.7, 157.0, and 118.3 mAhg^−1^, in comparison to the values of 180.1, 158.0, 139.3, 128.0, 115.8, and 97.0 mAhg^−1^ for the pristine Ti_3_C_2_T_x_ MXene, at the rate of 0.1, 0.5, 1, 2, and 5 C, respectively.

It is noted that MXenes are attractive candidates as anodes for LIBs. However, the performance of MXene-based composite anodes for LIBs is even more considerably studied in the literature because of their high-capacity anode materials, which manifest cyclability issues, such as very large volumetric expansions and contractions during electrochemical cycling. It is clear that the wide variety of composite MXenes, as listed in Table 3, offer considerable possibilities for the anodes of LIBs in the future.

## 5. Outlook

The number of applications for MXenes has recently increased; these include graphene as a 2D material, water purification, electromagnetic interference (EMI) shielding, biosensors, transparent conductive electrodes, and as electrodes for lithium-, sodium-, potassium-, and aluminum-ion batteries. It is worth mentioning that the MXene-based composite anode electrodes significantly improved the electrochemical performance of LIBs in previous studies, but this has been mostly achieved using laboratory-based coin cells. This is far from being useful in real-world applications because of a few reasons, such as the following: (i) the layer structure is prone to collapsing since MXene materials are extremely sensitive to oxygen and water; (ii) there are still a large number of unanswered scientific and technological questions regarding the MXene-based materials found in LIBs when used as an anode; and (iii) how MXene-based electrodes in LIBs function is currently unknown. As a result, there is still much to be done to implement MXene-based materials in LIBs. The purpose of this review was not to cover all the issues in practical applications. Instead, here we only focused on LIBs. MXenes can be considered promising anode materials for LIBs due to their superior mechanical properties, low Li^+^ diffusion resistance, and metal conductivity, as shown in Figure 15.

However, there are still several key issues that arise in the use of MXenes as anodes of LIBs, which need to be studied, as follows:
(a)Lithium compounds are used as cathode materials of commercial LIBs when graphite is used as the anode material. However, pure Li metal is used as a cathode material in most studies when MXene materials are used as the anode. Hence, it is necessary to identify the commercial-grade Li compound that is more suitable for the respective MXenes as anodes of LIBs.(b)At present, acid etching remains the primary way of obtaining MXenes; nevertheless, the approach has high risk and results in a low yield. As a result, there is a pressing need for safe, effective, high-quality, and environmentally friendly methods to synthesize MXenes that have a controlled number of layers, modifiable surface groups, enhanced layer spacing, and superior quality. To support the feasible large-scale commercial production of MXenes and MXene-based materials, it is also necessary to conduct extensive research on the generation mechanism and optimize the currently available synthesis techniques. In addition, the surface chemistry of these materials is mostly unknown, and thus must be studied more, before they can be used as anodes of LIBs.(c)At present, most synthesized MXenes are in powdered form, and thus need to be glued using binder materials and pasted on the current collector. The active materials of MXenes might lose their connection during the electrochemical cycling of LIBs, and mix with the electrolyte, which would lead to low practical energy density. (d)Suitable electrolytes that will not react with MXene materials during electrochemical cycling of LIBs need to be found. 

Success in addressing these issues should lead to the successful application of MXenes as anodes of LIBs at this very early stage of this type of research. However, this review has presented some noticeable achievements of various researchers all over the world regarding MXenes as anodes of LIBs. 

## 6. Conclusions

LIBs are the dominant energy storage technology of modern times. Due to its high capacitance, it can be inferred that the MXene is a promising material for use as the anode in an LIB. Different types of MXene synthesized by different techniques exhibit different properties. Monolayer MXenes exhibit better performance and cyclability compared to binder-free multilayer MXenes. They show comparatively higher specific capacity, even at high C rates. The double-transition metal MXenes deliver superior energy storage capacity compared to mono-transition MXenes due to the unique synergistic effects of bi-metals. The porous, doped, and pilled MXenes demonstrate a relatively high specific capacity at low C rates, and this phenomenon is anticipated due to the greater surface area, which is enabled by the presence of porosity, impurity, and defects in MXenes. However, the variations in specific capacity of porous, doped, and pilled MXenes at high C rates are yet to be investigated. Further experimental and computational research is required to completely understand the mechanisms of the underlying electrochemical processes and to optimize the associated performance to fully realize MXenes as anode materials for LIBs. We think that this review of recent breakthroughs in the synthesis and usage of MXene materials is a helpful starting point for the future development of these systems for use in LIB applications.

## Figures and Tables

**Figure 1 nanomaterials-14-00616-f001:**
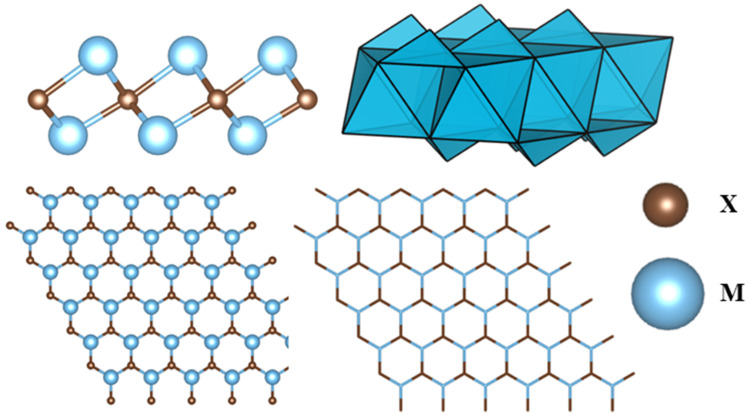
A typical M2X MXene structure without the terminates.

**Figure 2 nanomaterials-14-00616-f002:**
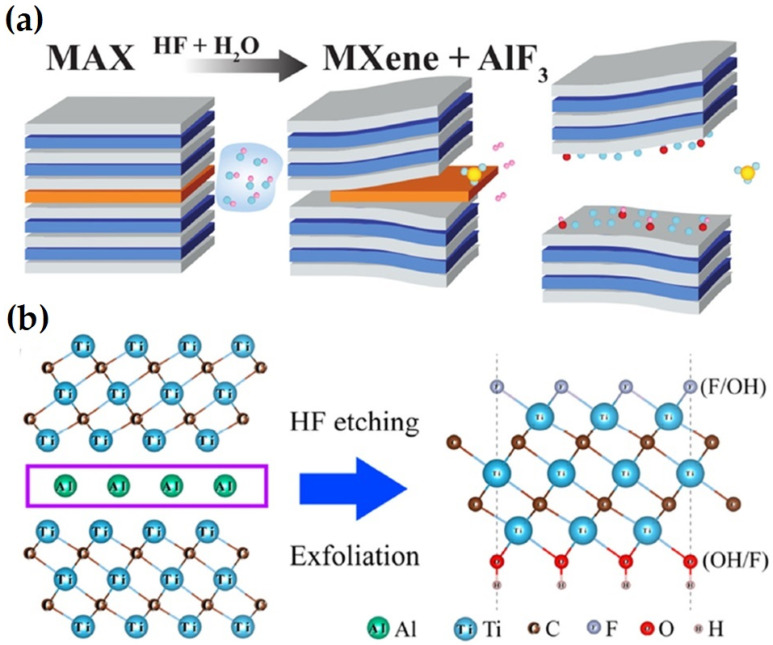
(**a**) Schematic diagram of the synthesis of MXene by HF etching (adapted from [39,40]). (**b**) The synthesis and structure diagram of Ti_3_C_2_T_x_ MXene (adapted from [41]).

**Figure 3 nanomaterials-14-00616-f003:**
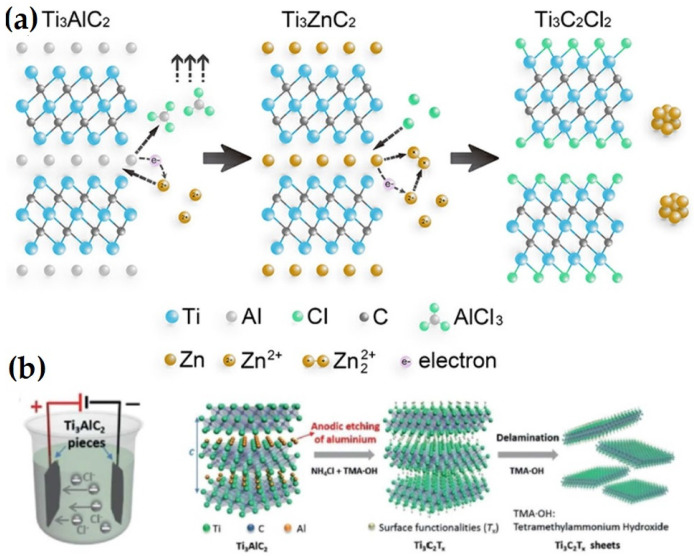
Schematic of the etching and delamination process. (**a**) Molten salt etching (adapted from [11]). (**b**) Electrochemical etching (adapted from [45]).

**Figure 4 nanomaterials-14-00616-f004:**
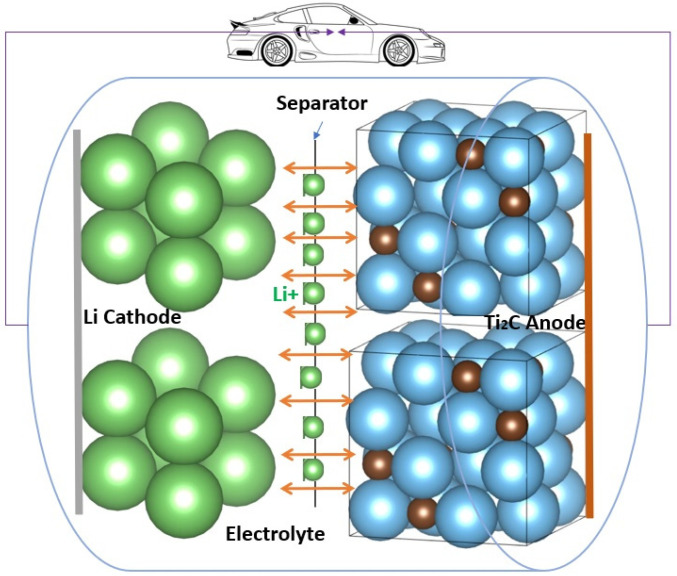
Schematic representation of MXene as an anode of LIBs.

**Figure 5 nanomaterials-14-00616-f005:**
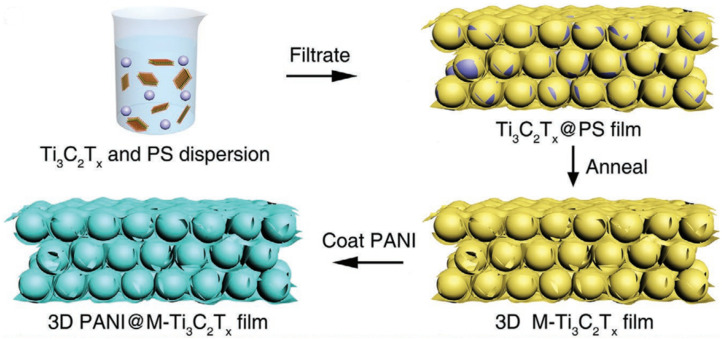
Schematic of 3D macroporous PANI@M-Ti_3_C_2_T_x_ frameworks using PS spheres as templates (adapted from [59]).

**Figure 6 nanomaterials-14-00616-f006:**
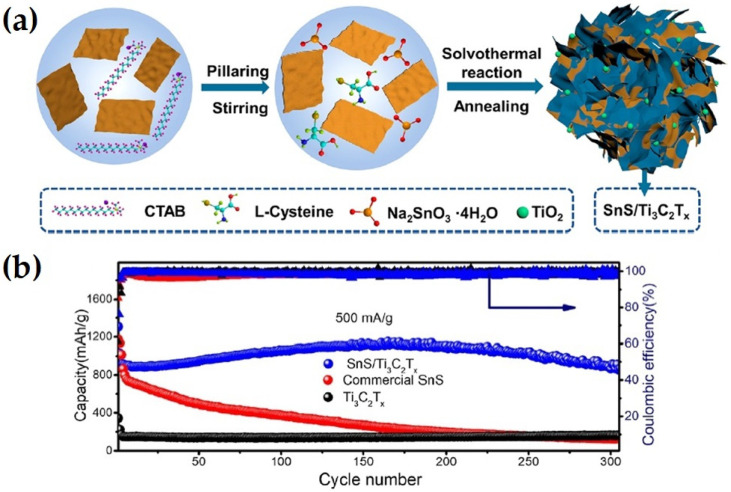
(**a**) Schematic illustration of the preparation process of SnS/Ti_3_C_2_Tx composites, and (**b**) cycling performance of SnS/Ti_3_C_2_T_x_ composites, pure Ti_3_C_2_T_x_, and commercial SnS at current rate of 500 mAg^−1^ (adapted from [62]).

**Figure 7 nanomaterials-14-00616-f007:**
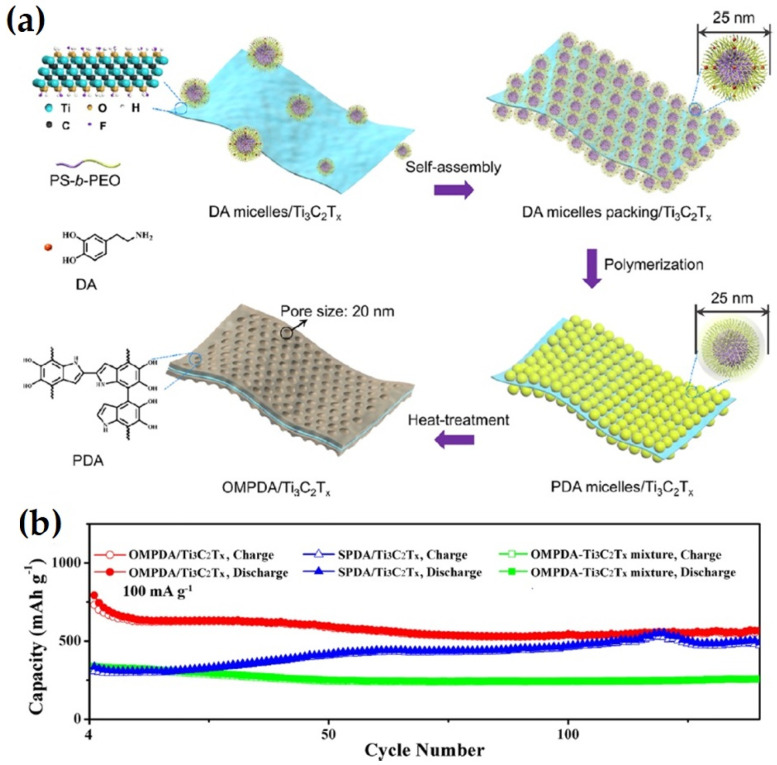
(**a**) Schematic drawing depicting the preparation steps of OMPDA/Ti_3_C_2_T_x_ composite, and (**b**) cycling stability of OMPDA/Ti_3_C_2_T_x_ anode (adapted from [65]).

**Figure 8 nanomaterials-14-00616-f008:**
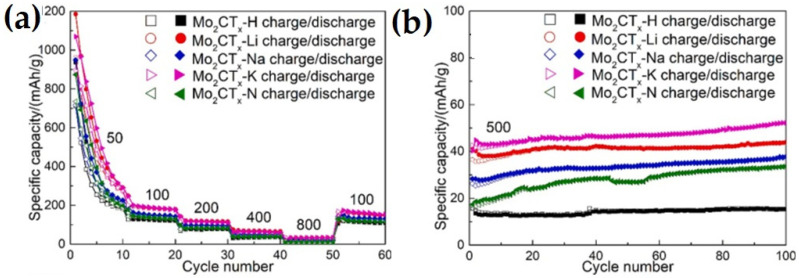
(**a**) Charge–discharge cycle performance curves of different samples at different current densities, (**b**) cycle stability curves of different samples at current density of 500 mAg^-1^ [73].

**Figure 9 nanomaterials-14-00616-f009:**
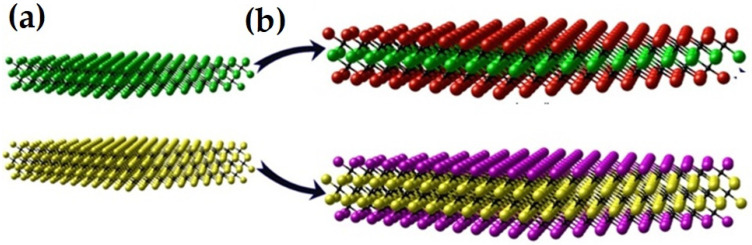
**Representation of structure of (a) solid solution of MXene and (b) ordered DTM. (adapted from [27])**.

**Figure 10 nanomaterials-14-00616-f010:**
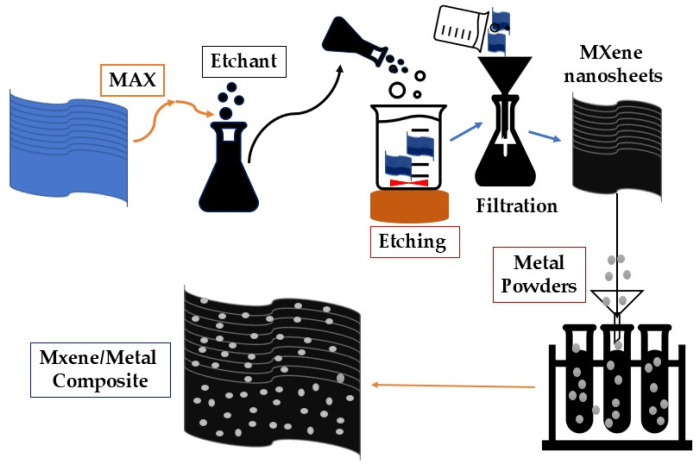
Schematic representation of the synthesis process of MXene/metal composite.

**Figure 11 nanomaterials-14-00616-f011:**
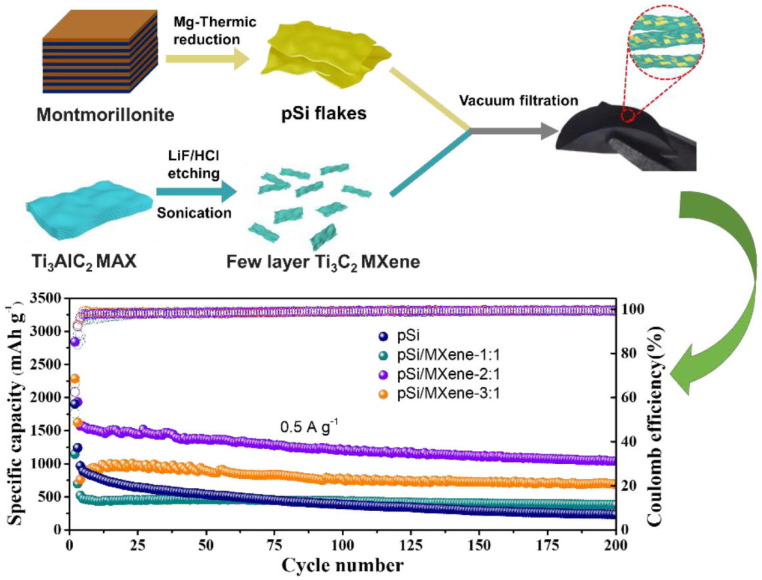
Schematic representation of synthesis and cyclic performances of different samples at 0.5 Ag^−1^ (adapted from [90]).

**Figure 12 nanomaterials-14-00616-f012:**
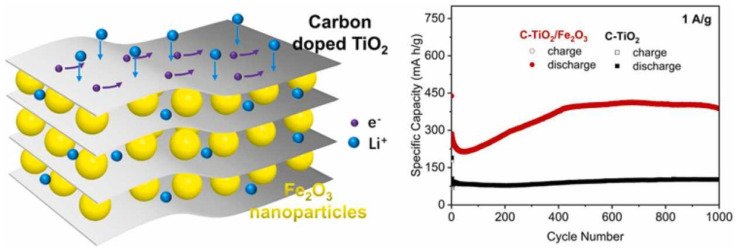
Schematic representation of C-TiO_2_/Fe_2_O_3_-Ti_3_C_2_ MXene with cyclic performance curves at 1 Ag^−1^ (adapted from [103]).

**Figure 13 nanomaterials-14-00616-f013:**
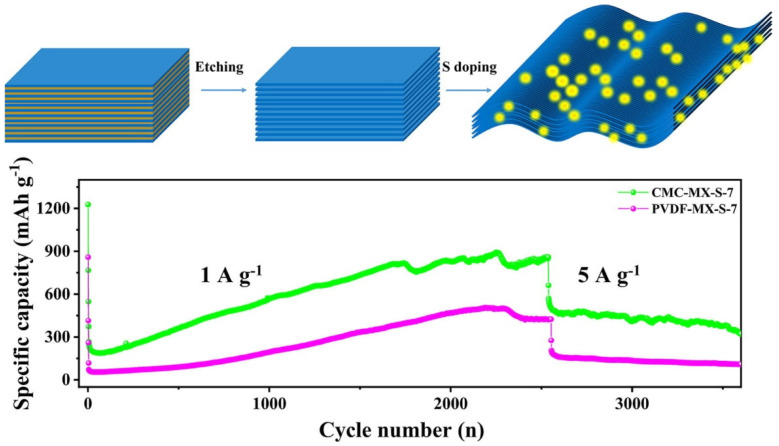
Cyclic performance curves for S-Ti_3_C_2_T_x_ MXene with CMC and PVDF binder (adapted from [118]).

**Figure 14 nanomaterials-14-00616-f014:**
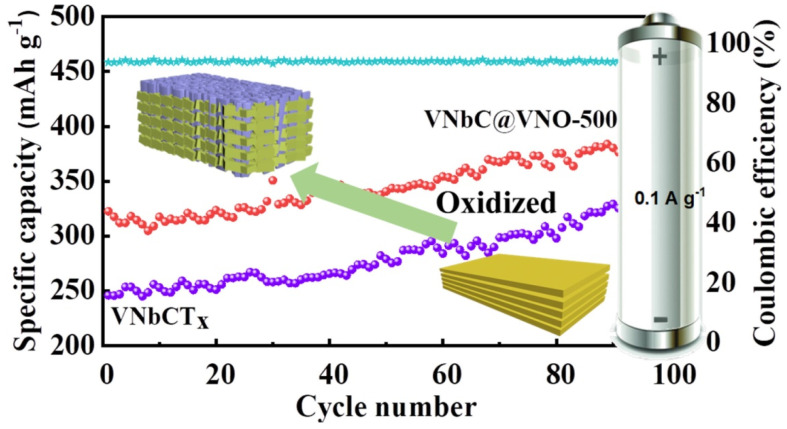
Cyclic stability curves of VNbC@VNO-500 and VNbCT_x_ MXene at 0.1 Ag^−1^ (adapted from [130]).

**Figure 15 nanomaterials-14-00616-f015:**
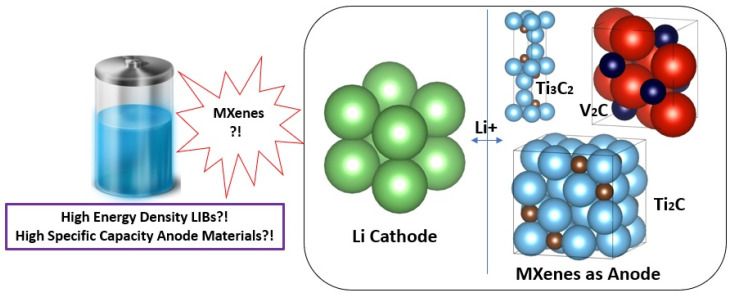
Hypothetical schematic representation of an LIB with an MXene as an anode.

**Table 1 nanomaterials-14-00616-t001:** Summary of the cyclic performance of 2D mono-transition metal MXenes as anodes of LIBs.

Anode Material	Etchant	Etching Time (h)	Discharge Capacity (mAhg^−1^) and Current Rate (Ag^−1^)	Number of Cycle	Ref.
Ti_2_C	10 wt% HF at RT	10	70 at 10 C	200	[53]
Ti_3_C_2_	49 wt% HF at 60 °C	24	118.7 at 1 C	75	[54]
Ti_3_C_2_	HCl + LiF at 35 °C	24	47.9 at 3 Ag^−1^	100	[55]
Ti_3_C_2_	NaOH		104.8 at 0.5 Ag^−1^	500	[56]
Partially etched Ti_3_C_2_	40 wt% HF at RT	0.5	160 at 1 C	100	[57]
Porous Ti_3_C_2_	HCl + LiF at 35 °C	24	220 at 1 Ag^−1^	3500	[58]
Ti_3_C_2_T_x_-400	40 wt% HF at 60 °C	48	126.4 at 1 C	100	[60]
Ti_3_C_2_T_x_-700	40 wt% HF at 60 °C	48	147.4 at 1 C	100	[60]
Scroll Ti_3_C_2_	HCl + LiF at 40 °C	24	~112 at 0.4 Ag^−1^	500	[61]
V_2_C	HCl + NaF at 90 °C	72	243 at 0.5 Ag^−1^	500	[67]
Nb_4_C_3_	49 wt% HF at RT	140	320 at 1 Ag^−1^	750	[68]
V_4_C_3_	40 wt% HF at 55 °C	96	125 at 1 Ag^−1^	300	[69]
Nb_2_CT_x_	Lewis acid at 750 °C	5	150 at 1 Ag^−1^	500	[70]
Nb_2_C	40 wt% HF at 60 °C	90	225 at 1 Ag^−1^	800	[71]
Partially etched V_2_CT_x_	40 wt% HF at RT	7 days	125 at 1 Ag^−1^	1000	[72]
d-Mo_2_CT_x_	HCl + KF at 180 °C	24	118.8 at 2 Ag^−1^	100	[73]
d-Hf_3_C_2_T_x_	35 wt% HF at RT	60	146 at 0.2 Ag^−1^	200	[74]
Ti_3_CNT_x_	HCl + LiF at 30 °C	12	300 at 0.5 Ag^−1^	1000	[75]
Mo_2_C	H_3_PO_4_	3-5	90 at 0.01 Ag^−1^	140	[78]

**Table 2 nanomaterials-14-00616-t002:** Summary of cyclic performance of the 2D DTM as an anode of LIBs.

Anode Material	Etchant	Etching Time (h)	Discharge Capacity (mAhg^−1^) and Current Rate (Ag^−1^)	Number of Cycle	Ref.
VNbCTx	50 wt% HF at 40 °C	48	520.5 at 0.1	100	[79]
Ti_2_NbC_2_T_x_	48 wt% HF at 50 °C	24	95.2 at 1	4000	[80]
(V_0_._7_Ti_0.3_)_2_C	LiF + HF at 90 °C	5	177 at 1	1000	[81]
(V_0_._5_Ti_0.5_)_2_C	LiF + HF at 90 °C	24	204.9 at 1	1000	[81]
(V_0_._3_Ti_0.7_)_2_C	LiF + HF at 90 °C	36	184.9 at 1	1000	[81]
Ti_x_Ta_(4−x)_C_3_	40 wt% HF at RT	72	459 at 0.5 C	200	[82]

**Table 3 nanomaterials-14-00616-t003:** Summary of cyclic performance of composite MXenes as anodes of LIBs.

Anode Material	Synthesis Process (Etchant)	Discharge Capacity (mAhg^−1^) and Current Rate (Ag^−1^)	Number of Cycles	Ref.
Ni(OH)_2_/d-Ti_3_C_2_	Hydrothermal (HCl + LiF)	372 at 1 Ag^−1^	1000	[84]
NiCo-LDH/Ti_3_C_2_	Hydrothermal (HCl + LiF)	562 at 5 Ag^−1^	800	[85]
NiFe-LDH/Ti_3_C_2_T_x_	Hydrothermal (HCl + LiF)	726.1 at 1 Ag^−1^	400	[86]
Ti_3_C_2_/Si	Orthosilicate hydrolysis and low-temp. reduction (HF)	973 at 1 Ag^−1^	800	[87]
Si/Ti_3_C_2_	Electrostatic self-assembly (HCl + LiF)	1654.8 at 1 Ag^−1^	300	[88]
Si/Ti_3_C_2_T_x_	Vacuum filtration (HCl + LiF)	1672 at 1 Ag^−1^	200	[89]
Porous Si/Ti_3_C_2_T_x_-2:1	Vacuum filtration (HCl + LiF)	555.5 at 5 Ag^−1^	500	[90]
Si-V_2_C	Ultrasonic mixing (NaF + HCl)	430 at 3 Ag^−1^	150	[91]
Mo_2_TiC_2_–Si-400	Pillaring and calcination (HCl + LiF)	108 at 1 Ag^−1^	500	[92]
SiO_2_/Ti_3_C_2_T_x_	Stober method spray drying (40 wt% HF)	635 at 1 Ag^−1^	200	[93]
Si-N-Ti_3_C_2_T_x_	Heat treatment (40 wt% HF)	760 at 3.2 Ag^−1^	900	[94]
SiO_x_-N-Ti_3_C_2_T_x_	Ball milling and annealing (40 wt% HF)	700 at 1 Ag^−1^	800	[95]
d-Si/G/V_2_C	48% HF + HCl	2003 at 1C	500	[96]
Fe-Ti_3_C_2_T_x_	HCl + LiF	418.8 at 0.2 Ag^−1^	500	[97]
Fe-Ti_3_C_2_	HF	310 at 5 Ag^−1^	850	[98]
(Fe-Ti) oxide/carbon/Ti_3_C_2_T_x_	Solvothermal, ultrasound hybridizing, and annealing	452.5 at 10 Ag^−1^	1200	[99]
N-Ti_3_C_2_/Fe_2_O_3_	Thermal decomposition (HCl + LiF)	549 at 2 Ag^−1^	400	[100]
β-FeOOH/Ti_3_C_2_T_x_	(HCl + LiF)	671 at 1 Ag^−1^	100	[101]
FTCN-Ti_3_C_2_	Sonochemical method (HCl + LiF)	1034 at 0.1 C	250	[102]
C-TiO_2_/Fe_2_O_3_-Ti_3_C_2_	Heat treatment	387.7 at 1 Ag^−1^	1000	[103]
Sn-Ti_3_C_2_T_x_	Molten salt reaction method (Lewis acid)	226.2 at 0.2 Ag^−1^	1000.	[104]
Sn^4+^/Ti_3_C_2_	Liquid phase immersion process (40 wt% HF)	544 at 0.5 Ag^−1^	200	[105]
V_2_CT_x_-SnO_2_	Ultrasound and annealing (NH_4_F + HCl)	274 at 8 Ag^−1^	200	[107]
Sn_4_P_3_-Ti_3_C_2_T_x_	Solvothermal phosphorization (50 wt% HF)	847 at 1 Ag^−1^	300	[108]
SnS/Ti_3_C_2_Tx	Solvothermal and annealing (HF)	866 at 0.5 Ag^−1^	300	[62]
SnS_2_/Sn_3_S_4_- Ti_3_C_2_T_x_	Solvothermal and calcination (45 wt% HF)	101.4 at 5 Ag^−1^	500	[109]
SnO_2_-Ti_2_C-C	Hydrothermal	763.18 at 2 Ag^−1^	500	[110]
Sn/SnO_x_-Ti_3_C_2_T_x_	Annealing (40 wt% HF)	594.2 at 0.05 Ag^−1^	200	[111]
5 wt% Ti_3_C_2_/TiO_2_	Hydrolysis (HCl + LiF)	180 at 0.1 C	100	[112]
Ti_3_C_2_/TiO_2_	Hydrothermal (46 wt% HF)	186 at 1 Ag^−1^	300	[113]
TiO_2_/Ti_2_C	Oxidation (10 wt% HF)	280 at 1 Ag^−1^	1000	[114]
Ti_3_C_2_/TiO_2_@f-MoS_2_	Hydrothermal and annealing (48 wt% HF)	403 at 2 Ag^−1^	1200	[115]
Ti_3_C_2_/LaF_3_	Heat treatment (HCl + LiF)	89.2at 1 Ag^−1^	50	[116]
Ti_3_C_2_/S		166.3 at 0.5 Ag^−1^	400	[117]
S-Ti_3_C_2_T_x_ (CMC)	Calcination and annealing (HCl + LiF)	858.2 at 5 Ag^−1^	3600	[118]
S-Ti_3_C_2_T_x_ (PVDF)	Calcination and annealing (HCl + LiF)	322.2 at 5 Ag^−1^	3600	[118]
Ti_3_C_2_/Ag	Self-reduction (40 wt% HF)	310 at 1 C	800	[119]
3D Ti_3_C_2_T_x_/Ag	HCl + LiF	310 at 3 Ag^−1^	2000	[120]
Li_3_VO_4_/Ti_3_C_2_	Sol–gel method (48 wt% HF)	146 at 5 C	1000	[121]
N-Nb_2_CT_x_	Hydrothermal (50 wt% HF)	238 at 5 C	100	[122]
MgH_2_/Ti_3_C_2_-60	HCl + LiF	328 at 2 Ag^−1^	50	[123]
GeO_x_/Ti_3_C_2_/PVDF(NMP)	One-pot method (HCl + LiF)	483 at 0.2 Ag^−1^	100	[124]
GeO_x_/Ti_3_C_2_/Li-PAA(DI-water).	One-pot method (HCl + LiF)	950 at 0.5 Ag^−1^	100	[124]
Ti_2_C/EMD		490 at 0.1 Ag^−1^	100	[125]
Bi_2_MoO_6_/Ti_3_C_2_T_x_-30%	Electrostatic self-assembly (HCl + LiF)	545.1 at 1 Ag^−1^	1000	[126]
Activated carbon-Ti_3_C_2_	Slurry casting method (HCl + LiF)	881.9 at 0.2 Ag^−1^	117	[127]
VO_2_-NTs/Ti_3_C_2_	Solvothermal self-assembly (HCl + LiF)	516 at 5 Ag^−1^	2000	[128]
TiNbC@NTO-500	Hydrothermal (HF)	261 at 1 Ag^−1^	500	[129]
VNbC@VNO-500	Partial oxidation (40 wt% HF)	323.9 at 1 Ag^−1^	1000	[130]
Na_2_Ti_3_O_7_/Ti_3_C_2_	Alkalization and oxidation (HCl + LiF)	158 at 4 Ag^−1^	1200	[131]
TiNb_2_O_7_/Ti_3_C_2_	Electrostatic self-assembly (HCl + LiF)	192.3 at 10 C	500	[132]
Ti_3_C_2_/CoS_2_	Hydrothermal (HCl + LiF)	~165 at 1 Ag^−1^	1000	[134]
MoS_2_/Ti_3_C_2_	Etching and solid-state sintering (HCl + LiF)	131.6 at 1 Ag^−1^	200	[135]
MoS_2_/Mo_2_TiC_2_T_x_	Sulfidation	509 at 0.1 Ag^−1^	100	[136]
Mo_3_Se_4_-Ti_3_C_2_T_x_	Hydrothermal (40 wt% HF)	1092.37 at 0.268 Ag^−1^	378	[137]
V_4_C_3_/MoS_2_/C	40 wt% HF	622.6 at 1 Ag^−1^	450	[138]
T-Ti_3_C_2_T_x_@C	HCl + LiF	337.3 at 2C	600	[140]
V_2_C@Co	HF	248.8 at 8 Ag^−1^	15,000	[141]
CoO/Ti_3_C_2_T_x_	Hydrothermal (40 wt% HF)	324 at 0.1 Ag^−1^	100	[142]
Co_3_O_4_/Ti_3_C_2_T_x_	HCl + LiF	307 at 5 C	1000	[143]
Co_3_O_4_@NGC/Ti_3_C_2_		830 at 1 Ag^−1^	500	[144]
Cu_2_O/Ti_2_C	Solvothermal (HCl + LiF)	143 at 1 Ag^−1^	250	[145]
Ti_3_C_2_/CNF	CVD	97 at 100 C	2900	[146]
Nb_2_C/CNT	CVD	430 at 2.5 C	300	[147]
d-Mo_2_C/CNT	Vacuum filtration (HCl + LiF)	76 at 10 Ag^−1^	1000	[148]
p-Ti_3_C_2_T_x_/CNT	Chemical etching method (HF)	~500 at 0.5 C	100	[14]
V_0.1_-Ti_3_C_2_T_x_	Microwave irradiation (HF)	72.9 at 3 C	1000	[149]
V_0.2_-Ti_3_C_2_T_x_	Microwave irradiation (HF)	92.4 at 3 C	1000	[149]
V_0.5_-Ti_3_C_2_T_x_	Microwave irradiation (HF)	98.3 at 3 C	1000	[149]

## Data Availability

All data are provided in the manuscript.

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
