# Peer review of "MXene as Promising Anode Material for High-Performance Lithium-Ion Batteries: A Comprehensive Review"

_nanomaterials, 2024, doi:10.3390/nano14070616_

Round 1
Reviewer 1 Report
Comments and Suggestions for Authors
In the review, authors summarize the recent progress of MXenes as promising anode material for Li-ion batteries. The review is well organized and timely. While, for better readership, several modifications are still needed as follows.
1. MXene assembly, like Ti3C2Tx nanoscrolls and hollow Ti3C2Tx submicro-tubes, and so on, should be further discussed, rather than just simple MXenes nanosheets. Additionally, typical synthetic methods are also needed in the revised version.
2. Besides, the self-supported MXenes should be summarized in detail as well.
3. More constructive perspectives should be proposed in the final section.
Comments on the Quality of English LanguageMinor modifications are needed
Author Response
Responses to reviewer’s comments
The authors would like to thank the reviewer for the valuable comments on the paper and have made changes in line with the comments.
In the review, authors summarize the recent progress of MXenes as promising anode material for Li-ion batteries. The review is well organized and timely. While, for better readership, several modifications are still needed as follows.
Comment 1: MXene assembly, like Ti3C2Tx nanoscrolls and hollow Ti3C2Tx submicro-tubes, and so on, should be further discussed, rather than just simple MXenes nanosheets. Additionally, typical synthetic methods are also needed in the revised version.
Response: This comment is highly appreciated, and the paragraph was provided in original manuscript in section “4.1 Mono Transition Metal MXene as Anode of LIBs”. However, there is only single research article we got during our study which is reported page 16-17 in Revised Manuscript as follows:
“Meng et al. [61] synthesized scrolled type Ti3C2 MXene by cold quenching in liquid nitrogen and compared the performance of Ti3C2 scroll with Ti3C2 sheet as anode of LIB for different current densities from 100 to 5000 mAg-1. The scroll type MXene exhibited excellent reversible capacities of 226 mAhg-1 (1st cycle), 155 mAhg-1 (22th cycle), 136 mAhg-1 (32th cycle), 113 mAhg-1 (42th cycle) and 89 mAhg-1 (62cycle) whereas, Ti3C2 MXene sheets achieved lower reversible capacities of 199 (1st cycle), 96 (22th cycle), 68 (32th cycle), 51 (42th cycle), 30 mAhg-1 (62th cycle) corresponding to the current densities of 100, 500, 1000, 2000 and 5000 mAg-1 respectively. The enhance electrochemical performance was reported due to more contact of electrolyte, greater inter layer spacing and shorter diffusion path for Li+ in the scroll structure. Furthermore, the Ti3C2Tx scrolls showed outstanding long-term cycling performance by retaining 81.6% of its initial capacity which was much better than the capacity retention of 63.3% of Ti3C2Tx sheets after 500 cycles at 400 mAg-1.”
2nd part of the comment 1: To our best knowledge, No research group reported Ti3C2Tx submicro-tubes as anode of LIBs till date. However, Sun, Xuan, et al. only reported as "Self-assembly construction of hollow Ti3C2Tx Submicro-Tubes towards efficient alkali metal ion storage." Chemical Engineering Journal 433 (2022): 134506. Hence, we are not including this paper in our present manuscript.
Comment 2: Besides, the self-supported MXenes should be summarized in detail as well.
Response: This comment is highly appreciated and following paragraphs are added in pages 17-21 of Revised Manuscript as follows:
“Zhang et al. [62] synthesized pillared SnS/Ti3C2Tx composites decorated with in situ formed TiO2 nanoparticles via solvothermal reaction and annealing treatment as shown in Fig. 6. The SnS/Ti3C2Tx composite exhibited high capacity of discharge 866 mAhg-1 at 500 mAg-1 current rate with 99% columbic efficiency which was better compared to those of commercial SnS and individual Ti3C2. The enhance electrochemical performance was ascribed to the pillar effect of Ti3C2Tx MXenes.
Figure 6. a) Schematic illustration of the preparation process of SnS/Ti3C2Tx composites and b) Cycling performance of SnS/Ti3C2Tx composites, pure Ti3C2Tx, and commercial SnS at current rate of 500 mAg-1. (Adapted from [62])
Recently Dai et al. synthesized self-supported and vertically aligned two dimensional (2D) heterostructures (V-MXene/V2O5) of rigid Ti3C2TX MXene and pliable vanadium pentoxide via an ice crystallization-induced strategy as anode of LIBs [63]. This thick V-MXene/V2O5 exhibited 472 and 300 mAh g–1 at a current rate of 0.2 A g–1, rate performance with 380 and 222 mAh g–1 retained at 5 A g–1, respectively, after 800 charge/discharge cycles. The enhance electrochemical performances was reported due to the vertical channels facilitated fast electron/ion transport within the entire electrode while the 3D MXene scaffold provided mechanical strength during Li+ insertion/deinsertion. Wang et al. synthesized Fe3O4@Ti3C2 MXene hybrid via a simple ultrasonication of Ti3C2 MXene and Fe3O4 nanoparticles and used as anode of LIBs [64]. The one of the compositions of Fe3O4@Ti3C2 hybrid exhibited high reversible capacities of 747.4 mA h g-1 at 1C after 1000 charge/discharge cycles. In addition, this anode material exhibited outstanding volumetric capacity up to 2038 mA h cm−3 at 1C due to the high compact density of the electrode of the prepared hybrid. Tao et al. fabricated mesoporous polydopamine (OMPDA)/Ti3C2Tx via in situ polymerization of dopamine on the surface of Ti3C2Tx via employing the PS-b-PEO block polymer as a soft template as shown in Fig. 7. This electrode exhibited average 1000 mAhg-1 of discharge capacity with 92 % columbic efficiency at 50 mAg-1 current rate after 200 cycles. The enhance electrochemical performance was attributed due the mesopores enhanced the overall capacity and reversibility of the reactions with Li+ [65].
Figure 7. a) Schematic drawing depicting the preparation steps of OMPDA/Ti3C2Tx composite and b) Cycling stability of OMPDA/Ti3C2Tx anode.(Adapted from [65])
Huan et al. [66] reported the Electrostatic Self-assembly of 0D-2D SnO2 Quantum Dots/Ti3C2Tx MXene hybrids as anode for LIBs. This electrode exhibited superior lithium storage properties with high capacity of 887.4 mAh g-1 at 50 mA g-1 current rate and a stable cycle performance of 659.8 mAh g-1 at 100 mA g-1 after 100 cycles with a capacity retention of 91%. It is noted that the enhance electrochemical performances was attributed due to the efficient pathways for fast transport of electrons and Li+ by Ti3C2Tx MXene. In addition, this Mxene materials buffered the volume change of SnO2 during Li insertion/deinsertion by confining SnO2 QDs between the nanosheets.”
Comment 3: More constructive perspectives should be proposed in the final section.
Response: We have revised section “5. Outlook” in Revised Manuscript in Pages-59-60 accordingly.
Reviewer 2 Report
Comments and Suggestions for Authors
Dear Editor and Authors:
MXene as Promising Anode Material for High-Performance Lithium-Ion Batteries: A Comprehensive Review
Overall, this work presents a comprehensive review of MXene for LIBs. It is a very relevant topic for both materials and energy storage scientists. It brings good information and it is well-written and narrated (although some improvements are needed). It is indeed, an interesting work and it deserves to be considered for publication. However, efforts are required to further improve the organization of this manuscript. Therefore, considering the high quality of Nanomaterials, I can not recommend the publication of this manuscript in its present form, and a moderate revision is needed.
Some other issues that need to be addressed are:
(1) Abstract: the abstract should be presented in a concise manner. Please bear in mind that a sharp abstract is central to article visibility and therefore to attract readers.
(2) Compared to previous works, address clearly what is the main contribution of this work topic to the empirical literature.
(3) The synthesis of MXenes should be better structured. Some schematics would be welcome. Also, address the main issues related to the MXene synthesis.
(4) 4. MXenes in LIBs. I believe the authors should better address the “role of MXenes surface and its modification on electrode properties and battery metrics”
(5) A more structured discussion about the role of the porosity of MXenes on the battery metrics should be addressed.
(6) 4.3. Composite MXene as the anode of LIBs: This section is very long and it just contains the description of works. I believe the authors should also try to be more critical. Also, some schematics would be welcome.
(7) The authors are suggested to add their own opinions on the recent progress and outlook in the main text, rather than just retell the results from other papers.
(8) The main challenges (if any) for the use of MXenes in LIBs should be pointed out for better guidance of the readers.
Author Response
Responses to reviewer’s comments
The authors would like to thank the reviewer for the valuable comments on the paper and have made changes in line with the comments.
Overall, this work presents a comprehensive review of MXene for LIBs. It is a very relevant topic for both materials and energy storage scientists. It brings good information and it is well-written and narrated (although some improvements are needed). It is indeed, an interesting work and it deserves to be considered for publication. However, efforts are required to further improve the organization of this manuscript. Therefore, considering the high quality of Nanomaterials, I can not recommend the publication of this manuscript in its present form, and a moderate revision is needed. Some other issues that need to be addressed are:
Comment 1: Abstract: the abstract should be presented in a concise manner. Please bear in mind that a sharp abstract is central to article visibility and therefore to attract readers.
Response: We do appreciate this comment and edited the Abstract in Revised Manuscript as follows:
“Broad adoption has already been started of MXenes materials in various energy storage technologies such as super-capacitors and batteries due to the increasing versatility of the preparation methods as well as the ongoing discovery of new members. The essential requirements for an excellent anode material for Lithium-ion batteries (LIBs) are high safety, minimal volume expansion during the lithiation/de-lithiation process, high cyclic stability, and high Li+ storage capability. However, most of the anode materials for LIBs, such as graphite, SnO2, Si, Al, Li4Ti5O12, etc., have at least one issue. Hence, creating novel anode materials continues to be difficult. Few MXenes have been investigated experimentally as anode of LIBs till date due to their distinct active voltage windows, large power capabilities, and longer cyclic life. The objective of this review paper is to provide an overview of the synthesis and characterization characteristics of the MXenes as anode materials of LIBs, including its discharge/charge capacity, rate performance, and cycle ability. In addition, a summary of potential outlook of developments of these materials as anode in the future is suggested.”
Comment 2: Compared to previous works, address clearly what is the main contribution of this work topic to the empirical literature.
Response: We appreciate this comment and edited accordingly in sections “5. Outlook” and “6. Conclusion”. However, MXene material is new material for application as electrode of various batteries. Researcher are trying their best to resolve some of the issues which are essential for practical applications in LIBs. Hence, it is necessary to know present researcher which is going on in this field in a comprehensive manner with research gap and others effort to improve this new system.
Our manuscript did the same. It has presented comprehensive summary of MXene as anode of LIBs with some research directions which are explained in sections “5. Outlook” and “6. Conclusion” in Revised Manuscript.
Comment 3: The synthesis of MXenes should be better structured. Some schematics would be welcome. Also, address the main issues related to the MXene synthesis.
Response: We appreciate this comment and added the following figures in Revised Manuscript:
“Figure 2. (a) Schematic diagram of the synthesis of MXene by HF etching (Adapted from [39,40] ) and b) The synthesis and structure diagram of Ti3C2Tx Mxene. (Adapted from [41])
Figure 3. Schematic of the etching and delamination process. (a) Molten salts etching. (Adapted from [11]) and (b) Electrochemical etching. (Adapted from [45])”
Comment 4. MXenes in LIBs. I believe the authors should better address the “role of MXenes surface and its modification on electrode properties and battery metrics”
Response: We do appreciate this comment. However, the surface morphology of the same MXene materials is exhibiting different structures due to various synthesis parameters and other unknown parameters. Hence, at this stage of the research in this field, it is tough to get enough information to compare and put perspective on this. Hence, we deliberately overlooked this part.
Comment 5. A more structured discussion about the role of the porosity of MXenes on the battery metrics should be addressed.
Response: We appreciate this comment and answered this comment in the response of comment 4.
Comment 6: Composite MXene as the anode of LIBs: This section is very long, and it just contains the description of works. I believe the authors should also try to be more critical. Also, some schematics would be welcome.
Response: We have edited the Section “4.3 Composite MXene as anode of LIBs” in the revised manuscript and added the following figures:
“Figure 10. Schematic representation of synthesis process of MXene/Metal composite.
Figure 11. Schematic representation of synthesis and cyclic performances of different samples at 0.5 Ag-1. (Adapted from [90])
Figure 12. Schematic representation of C-TiO2/Fe2O3-Ti3C2 MXene with cyclic performance curves at 1 Ag-1. (Adapted from[103])
Figure 13. Cyclic performance curves for S-Ti3C2Tx MXene with CMC and PVDF binder (Adapted from [118])
Figure 14. Cyclic stability curves of VNbC@VNO-500 and VNbCTx MXene at 0.1 Ag-1. (Adapted from [130])”
Comment 7: The authors are suggested to add their own opinions on the recent progress and outlook in the main text, rather than just retell the results from other papers.
Response: We edited the Revised Manuscript accordingly and addressed this comment in sections “5. Outlook” and “6. Conclusion” in Revised Manuscript.
Comment 8: The main challenges (if any) for the use of MXenes in LIBs should be pointed out for better guidance of the readers.
Response: We have addressed this comment in sections “5. Outlook” and “6. Conclusion” in Revised Manuscript.
Round 2
Reviewer 2 Report
Comments and Suggestions for Authors
No more comments. This version is now suitable for publication.